# Theory of optimal balance predicts and explains the amplitude and decay time of synaptic inhibition

Jaekyung K. Kim[1] & Christopher D. Fiorillo[1]

Synaptic inhibition counterbalances excitation, but it is not known what constitutes optimal inhibition. We previously proposed that perfect balance is achieved when the peak of an excitatory postsynaptic potential (EPSP) is exactly at spike threshold, so that the slightest variation in excitation determines whether a spike is generated. Using simulations, we show that the optimal inhibitory postsynaptic conductance (IPSG) increases in amplitude and decay rate as synaptic excitation increases from 1 to 800 Hz. As further proposed by theory, we show that optimal IPSG parameters can be learned through anti-Hebbian rules. Finally, we compare our theoretical optima to published experimental data from 21 types of neurons, in which rates of synaptic excitation and IPSG decay times vary by factors of about 100 (5–600 Hz) and 50 (1–50 ms), respectively. From an infinite range of possible decay times, theory predicted experimental decay times within less than a factor of 2. Across a distinct set of 15 types of neuron recorded *in vivo*, theory predicted the amplitude of synaptic inhibition within a factor of 1.7. Thus, the theory can explain biophysical quantities from first principles.

[1] Department of Bio and Brain Engineering, Program in Brain and Cognitive Engineering, KAIST, Daejeon 34141, Republic of Korea. Correspondence and requests for materials should be addressed to C.D.F. (email: fiorillo@kaist.ac.kr).

Synaptic inhibition approximately counterbalances excitation on a millisecond timescale[1–15]. Although balanced inhibition has advantages[16–20], it is not known what constitutes optimal balance. We recently proposed that perfect homoeostatic balance is momentarily achieved when the peak of an EPSP is exactly at spike threshold, so that whether a spike does or does not occur will depend on the slightest variation in the amplitude of the excitatory postsynaptic conductance (EPSG)[21]. This would maximize the information that 'spike' or 'no spike' conveys about EPSG amplitude.

If correct, the theory should explain and predict biophysical properties. There is a wealth of data showing that IPSG mediated by GABA$_A$ and glycine receptors have decay time constants that vary from 1 to at least 50 ms across diverse types of neurons, and a lot is known about the biophysical basis of this diversity[22]. But why should a specific synapse have the specific IPSG decay time that it does? Why should decay times vary across synapses? A good theory should explain the specific decay times and their variability from fundamental principles.

The proposed goal of homoeostatic balance is to maximize the causal and informational link between EPSG amplitude and spike generation. In other words, a spike should accurately 'measure' EPSG amplitude. Like any other measurement, this requires comparison to a reference. A balance scale measures weight in almost perfect analogy to the way that membrane voltage and spikes measure EPSG (Fig. 1a,b). The angle of the arm (membrane voltage) depends on the difference between the new and unknown weight of interest on one side (EPSG) and a known reference weight on the other (IPSG and all other factors that counteract depolarization). If the balance scale has a binary output (spike or no spike), that output indicates only whether the new weight (EPSG) is more (spike) or less (no spike) than the reference (IPSG). Measurement is most accurate when perfect balance is attained (EPSP peak at spike threshold), and this is the homoeostatic ideal.

Balance is difficult to achieve because EPSG vary dynamically across a large range of amplitudes and on a timescale of a millisecond or less. Balance requires that inhibition is dynamically adjusted to predict EPSG amplitude, analogous to continually adjusting the reference weight in anticipation of a new weight being placed on the balance scale. We have therefore referred to this process as 'predictive homoeostasis'[21], and it is related to principles of 'predictive coding'[23–25]. It is 'homoeostatic' insofar as the parameters of inhibition (synaptic strength, decay time and so on) are adjusted through negative feedback (for example, anti-Hebbian plasticity) to drive membrane excitability towards an intermediate target (optimal balance), and it is 'predictive' insofar as these parameters are determined in advance to counterbalance the expected future amplitude of excitation.

A useful example is a thalamocortical neuron of the lateral geniculate nucleus (LGN) recorded during viewing of a movie (Fig. 1c)[26]. As expected for effective homoeostatic balance, many of the large retinogeniculate EPSPs cause spikes and many do not ($\sim$50% on average across neurons)[27–30], indicating that spike generation is highly sensitive to the precise amplitude of retinogeniculate EPSG. Retinogeniculate EPSG are the proximate cause of all spikes, which signify evidence against light in this OFF-type neuron (Fig. 1c). In addition, there are two types of IPSG that convey evidence of opposite polarity (Fig. 1d)[26,31–33]. 'Homoeostatic IPSG' convey evidence of the same polarity as EPSG (evidence for EPSG amplitude, which is evidence against light in this example), whereas 'opponent IPSG' convey evidence against EPSG amplitude. Opponent IPSG are anti-correlated (out of phase) with EPSG (since simultaneous evidence for and against light in the same small receptive field is unusual) and thus they tend to cause hyperpolarization (Fig. 1c,d)[26,34]. In contrast, homoeostatic IPSG occur at nearly the same time as EPSG, and thus suppress EPSP amplitude. Homoeostatic IPSG are consistently delayed from EPSG onset by

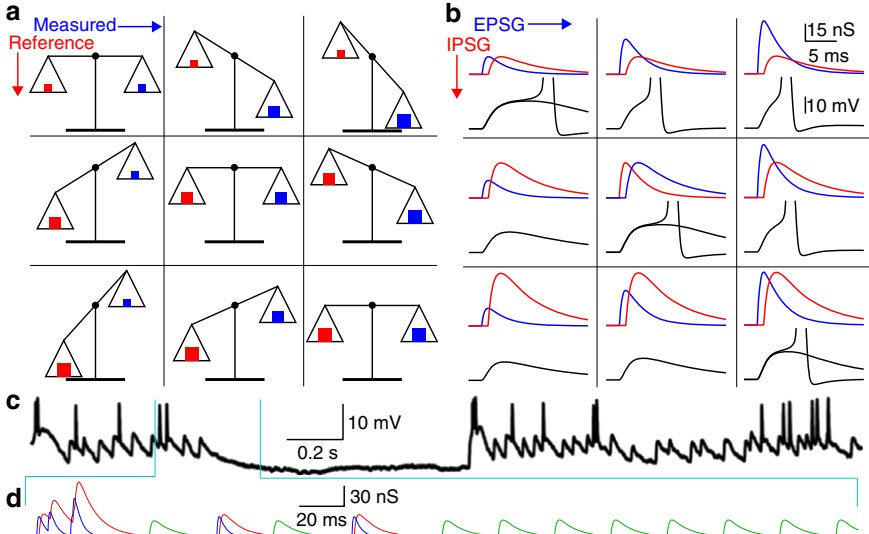

**Figure 1 | Illustration of theory by analogy to a balance scale for measuring weight.** (**a**) Each of the nine configurations of the balance scale are analogous to the corresponding combinations of EPSG, IPSG and membrane voltage in **b**. When EPSG amplitude exceeds the counterbalance provided by the IPSG and all other factors that counteract depolarization, an AP is generated (depolarized phase of AP not shown). Perfect homoeostatic balance is achieved when EPSP peak is exactly at AP threshold (diagonal from top left to bottom right; since an AP is equally likely to occur or not occur, we have shown overlaid traces with and without AP). The illustrated EPSG and IPSG amplitudes were chosen only for convenience. (**c**) Membrane voltage in a thalamocortical neuron in cat LGN during viewing of a naturalistic movie (reprinted from ref. 26, copyright 2007, with permission from Elsevier). (**d**) Estimates of the amplitude and timing of retinogeniculate EPSG (blue), homoeostatic IPSG (red) and opponent IPSG (green) that contribute to the membrane voltage in **c** during the period indicated (cyan lines). Estimates were based on experimental observations (Supplementary Tables 1 and 2)[26,28,29,31], but our primary purpose here is only to illustrate that homoeostatic and opponent IPSG are 'in' and 'out' of phase with EPSG, respectively.

1 ms in LGN (due to a dendrodendritic synapse)[31]. Despite their opposing information content, homoeostatic and opponent IPSG have similar amplitudes and decay times in LGN[31].

Our present focus is homoeostatic IPSG, which are likely to be more prevalent than opponent IPSG since not all neurons have an 'opponent neuron'. Through computer simulations of a simple and generic model neuron, we identified the IPSG decay time constant ($\tau$) and mean peak amplitude ($I$) that are optimal given a particular temporal pattern of EPSG. We found that these optimal IPSG closely match experimental data.

## Results

**Single-compartment model.** We began with a single-compartment model having only a constant 'leak' conductance ($G_L$, reversal at $-70$ mV), and IPSG that always followed EPSG by 1.0 ms. In initial simulations, our model neuron did not have action potentials (AP), and 'spike threshold' was $-50$ mV. Although conductance amplitudes naturally vary from one synaptic event to another, our interest was the contribution of postsynaptic strength, which changes more slowly. Thus unitary mean peak amplitudes ($I$ and $E$) were constant over time in each simulation. We usually refer to the ratio of amplitudes '$I/E$,' where '$I$' was the parameter of interest and '$E$' a constant (30 nS standard).

For each EPSG we measured 'distance from optimality' (Fig. 2a), which we called a 'residual' (Methods)[21]. For a given ensemble of EPSG, optimal IPSG parameters were taken to be those that minimized the mean squared residual (MSR).

**Fixed intervals between pairs of EPSG.** We start with the simplistic but illustrative example of a pair of EPSG separated by a fixed interval, with inhibition provided only by $G_L$. $G_L$ corresponds to the special case of an IPSG in which $\tau$ approaches infinity and amplitude approaches zero. Since excitation can be approximately counterbalanced by $G_L$ alone, we need to understand why neurons have 'fast IPSG'.

MSR is naturally a U-shaped function of $G_L$, with $G_L$ that minimizes MSR being optimal (Fig. 2b). For long intervals (no temporal summation), optimal $G_L$ causes the peak of each EPSP to be exactly at threshold (MSR = 0). For short intervals, temporal summation of EPSP requires greater $G_L$ to counterbalance excitation, but optimal $G_L$ cannot eliminate residuals (Fig. 2a,b). The first EPSP will be sub-threshold (too much inhibition) and the second super-threshold (too little inhibition). A dynamic inhibitory conductance that is stronger at the time of the second EPSG could better balance excitation. Indeed, adding an IPSG further reduced MSR (compare minima in Fig. 2b,c). For long intervals with no temporal summation, synaptic and leak conductances could fully substitute for one another, but for short intervals a single combination was optimal (Fig. 2d).

**Optimal IPSG are larger and faster at higher EPSG frequency.** We identified optimal IPSG parameters given inter-EPSG intervals randomly drawn from geometric distributions (time unit of 1.0 ms) with mean rates of 1–800 Hz. With EPSG of 30 nS, $G_L$ of 10 nS was near optimal (based on systematically varying $G_L$, $I$, $\tau$ and frequency), and this combination defined our 'standard model'.

Although many combinations of $I/E$ and $\tau$ approximately balanced excitation, there was one combination that was optimal for each EPSG frequency (Fig. 3a–d). MSR was a smooth function of $\tau$ over 1–120 ms, and optimal $G_L$ resulted in MSR just slightly greater than $\tau$ of 120 ms (Fig. 3d). Therefore we are quite confident that we identified the one optimal $\tau$ from the infinite range of possibilities.

We illustrate two $\tau$ (2.2 and 26 ms) that were optimal at 400 and 5 Hz, respectively (with $I/E$ optimized in all four cases) (Fig. 4). Relative to optimal and slow $\tau$ at 5 Hz, excessively fast decay resulted in sub-threshold EPSPs that tended to be smaller (since $I/E$ was too high), and super-threshold EPSPs that tended to be larger (since faster IPSG decay resulted in slower EPSP decay and thus greater temporal summation). At 400 Hz, $\tau$ that

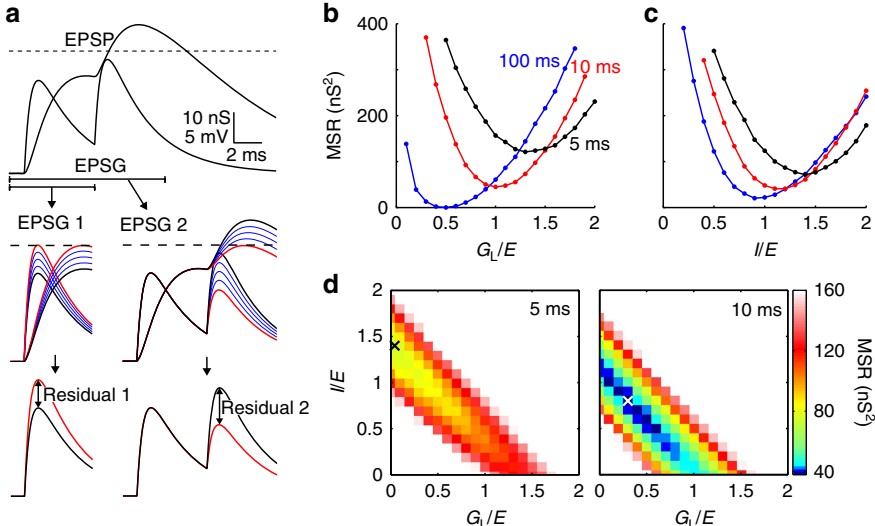

**Figure 2 | Optimal inhibition in relation to EPSG interval.** (**a**) Method of measuring 'residuals'. Top, two EPSG and the resulting EPSP. Dashed line indicates $-50$ mV, the 'spike threshold' in these simulations without AP. Middle, 'test EPSG' (blue) were injected at the time of 'real EPSG' (black) to find the 'threshold EPSG' (red) that caused an EPSP with its peak to be nearest to threshold (in other words, EPSG amplitude was varied, but this 'test variation' was only to find spike threshold, and had no influence on 'real' voltage at the time of subsequent EPSG). Bottom, the residual was the difference in peak amplitudes between threshold and real EPSG. The residual depended on current and past but not future synaptic events; thus residual 1 in this example (left) was found in simulations in which EPSG 2 did not occur. (**b**) MSR as a function of the ratio of $G_L$ to $E$ (30 nS) for two EPSG separated by 5 (black), 10 (red) and 100 ms (blue) in the absence of synaptic inhibition ($I/E = 0$). (**c**) As in **b**, but showing MSR as a function of $I/E$ ($\tau = 10$ ms, $G_L = 0.1$ nS). (**d**) Heat plots of MSR as a function of both $G_L/E$ and $I/E$ for inter-EPSG intervals of 5 (left) and 10 ms (right). 'X' indicates minimum MSR.

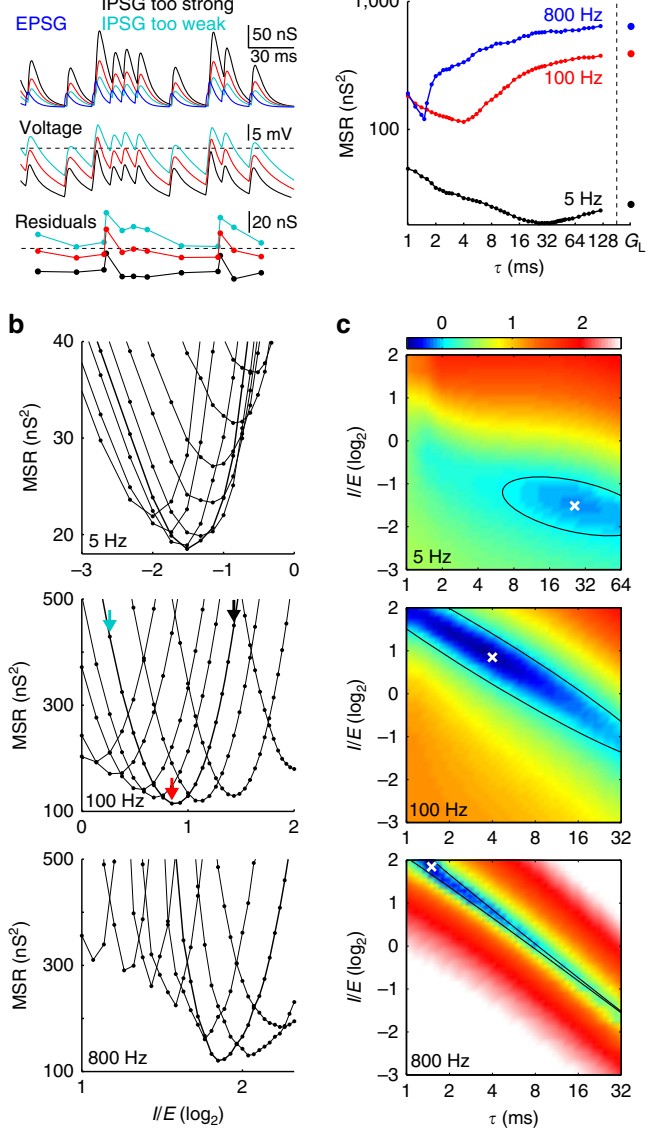

**Figure 3 | Finding optimal IPSG with random EPSG intervals.**
(**a**) Conductances (top), voltages (middle) and residuals (bottom) for the case that I/E was too strong (black), too weak (cyan) and near optimal (red) for EPSG (blue) at 100 Hz. Dashed lines indicate −50 mV (spike threshold) and residual of zero. The sign of residuals was chosen so that positive residuals corresponded to EPSP peaks more positive than −50 mV. (**b**) For each value of τ, MSR as a function of I/E for EPSG at 5, 100 and 800 Hz (top to bottom). Thicker lines indicate τ that minimized MSR. Arrows indicate values of I/E shown in **a**. (**c**) Heat plots of MSR (same data as **b**). For each EPSG frequency, MSR was divisively normalized by the MSR obtained with no IPSG and only optimal $G_L$ (see below). Black ellipses indicate IPSG parameter values that resulted in MSR equivalent to optimal $G_L$ in the absence of IPSG. 'X' indicates minimal MSR. (**d**) MSR as a function of τ for EPSG at 5 (black), 100 (red) and 800 (blue) Hz. I/E was optimized for each τ, and thus each data point corresponds to the minimum of one function in **b**. At far right is MSR with no IPSG and only optimal $G_L$ (25.6, 389.4 and 638.0 nS at 5, 100 and 800 Hz, respectively), which corresponds to an IPSG with infinitely slow τ.

was too slow resulted in IPSG that were substantially out of phase with EPSG, resulting in periods of excessive depolarization and hyperpolarization (Fig. 4a, right). Interestingly, EPSP had earlier peak latencies and faster decay with optimal versus suboptimal τ

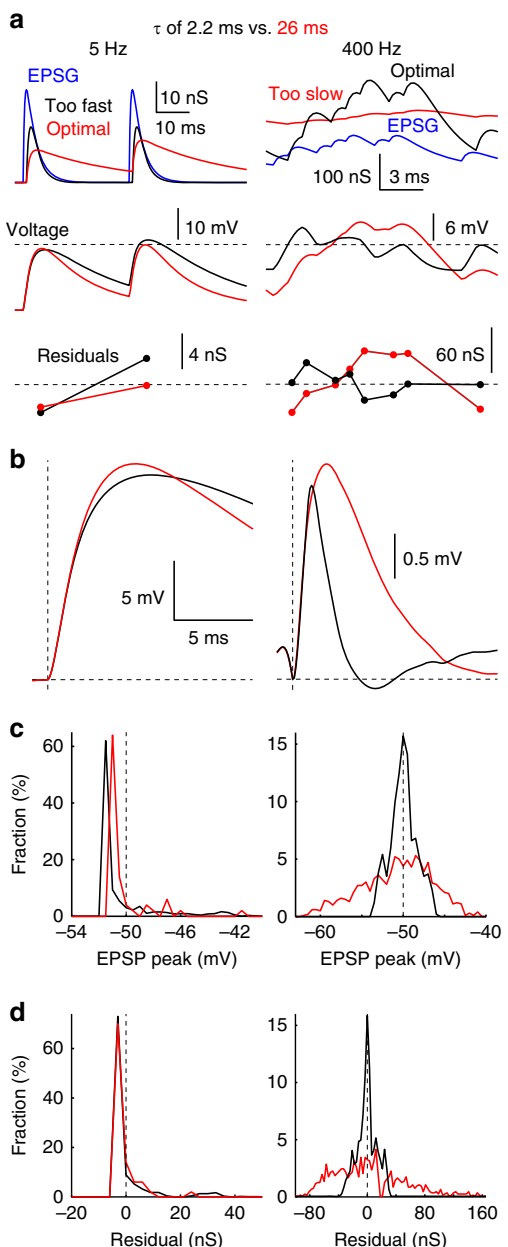

**Figure 4 | Comparison of optimal and suboptimal IPSG decay times.**
(**a–d**) τ of 2.2 (black) and 26 ms (red) at EPSG frequencies of 5 (left) and 400 Hz (right). I/E was optimized in all four cases, whereas τ of 26 and 2.2 ms were optimal at 5 and 400 Hz, respectively. (**a**) Examples of EPSG and IPSG (top), voltage (middle), and residuals (bottom). (**b**) EPSG-triggered average EPSPs. (**c**) Histograms of EPSP peaks. Dashed lines indicate 'spike threshold' of −50 mV; EPSP peaks positive of −50 mV were designated as positive residuals. (**d**) Histograms of residuals.

at both 5 and 400 Hz, demonstrating that faster voltage dynamics are favoured by either fast or slow IPSG depending on EPSG frequency (Fig. 4b). This helps to explain how optimal τ was near optimal in minimizing spike latency and variability (see below).

As EPSG rate increased from 1 to 800 Hz, optimal I/E increased and τ decreased (Fig. 5a,b). Even with optimal IPSG parameters, homoeostasis was not as well maintained at high frequencies (MSR was greater), although it was maintained better with optimal IPSG than with only optimal leak conductance (Fig. 5c).

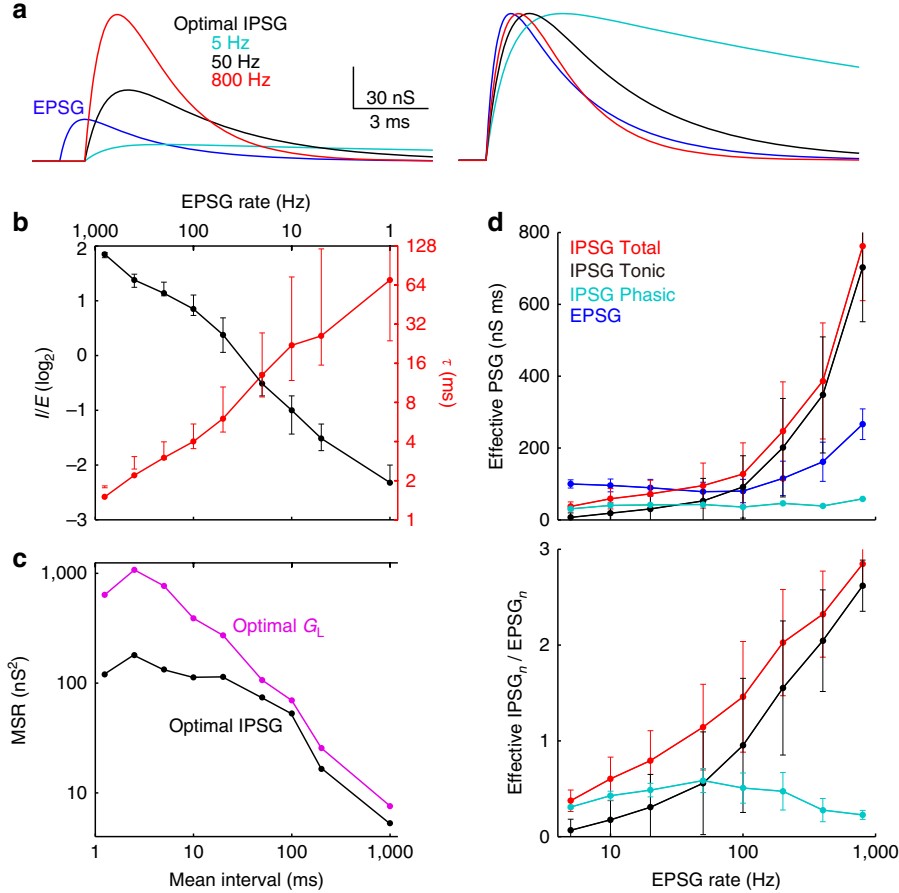

**Figure 5 | Optimal IPSG parameters depend on EPSG frequency.** (**a**) Optimal IPSG for EPSG at 5 (black), 50 (green) and 800 Hz (red). Right, same as left but with EPSG (blue) and IPSG onsets aligned and peaks normalized. (**b**) Optimal $I/E$ (black) and $\tau$ (red) as a function of mean EPSG interval (bottom axis) and frequency (top axis), with log–log coordinates. Bars indicate the parameters necessary to increase MSR by 10% above the minimum achieved by optimal parameters. (**c**) MSR as a function of frequency with optimal IPSG (black) and $G_L$ (magenta). (**d**) Tonic and phasic postsynaptic conductances (PSG). To estimate the 'effective PSG' that influenced the causal link between EPSG and spike generation, we found the mean spike latency at each frequency (5.0–1.7 ms at 5–800 Hz) and we integrated each PSG from EPSG onset to mean latency. The effective PSG was averaged (mean ± s.d.) across all synaptic events (top), or the ratio of effective IPSG to EPSG at the time of each event was averaged across all synaptic events (bottom). The 'total IPSG', but not EPSG, was split into tonic and phasic components.

**Tonic and phasic components of IPSG.** Whether $EPSG_n$ causes a spike will depend on both the 'phasic' $IPSG_n$, as well as the 'tonic' component contributed by summation of $IPSG_{n-1}$, $IPSG_{n-2}$ and so on. We estimated these components by integrating conductances from EPSG onset to mean spike time (5.0–1.7 ms at 5–800 Hz). The tonic IPSG increased with frequency and was larger than the phasic IPSG at 100 Hz and above (Fig. 5d), despite optimal IPSG being larger and decaying faster at higher frequencies. Thus one effect of optimization is to counteract the natural tendency of the tonic component to overwhelm the phasic component as frequency increases.

In our standard model (above), both tonic and phasic components made significant contributions to suppressing spikes, and thus optimization of IPSG depended on both components. However, phasic inhibition was reduced in a model with much greater membrane conductance, especially at high frequencies (due to decreased spike latency). This flattened the relations of optimal $I/E$ and $\tau$ to frequency (see below). The relations became flatter still when we eliminated all phasic inhibition in this high-conductance model by increasing the delay from EPSG to IPSG onset (E–I delay) from 1.0 to 2.0 ms (optimal $\tau$ was 7.0, 3.0 and 6.0 ms at 5, 100 and 800 Hz; optimal $I/E$ was 0.9 at 800 Hz, less than any other frequency).

**Influence of additional parameters.** We modified our standard model to examine a variety of additional factors (Figs 5–7; Supplementary Figs 1–2). Although each was influential, none fundamentally altered the dependence of optimal $I/E$ and $\tau$ on EPSG frequency that was observed with our standard model (Fig. 5b,c; Supplementary Results).

Our standard model was simplified in using unitary EPSG and IPSG that were of constant amplitude over time. However, variability *in vivo* is much less than one would expect given typical conditions in brain slices[35]. Under naturalistic conditions at the calyx of Held, EPSC at 40 Hz had a coefficient of variation (CV) of 8% (ref. 36). We performed additional simulations with unitary synaptic conductance amplitudes having the same mean (30 nS) but CVs of 15 and 79%, with normal and log-normal distributions, respectively (Fig. 6a,b) (in the latter case, the standard deviation corresponded to 15–60 nS, a factor of 2 from the mean). EPSG and IPSG varied independently (although we would expect moderate covariance under natural conditions). Relative to our standard model, the lesser variance had virtually no effect on optimal IPSG parameters, whereas larger variance with a skewed distribution decreased the amplitude and more than doubled the decay time at all but the highest frequencies (Fig. 6c). MSR increased with variance, as expected (Fig. 6d).

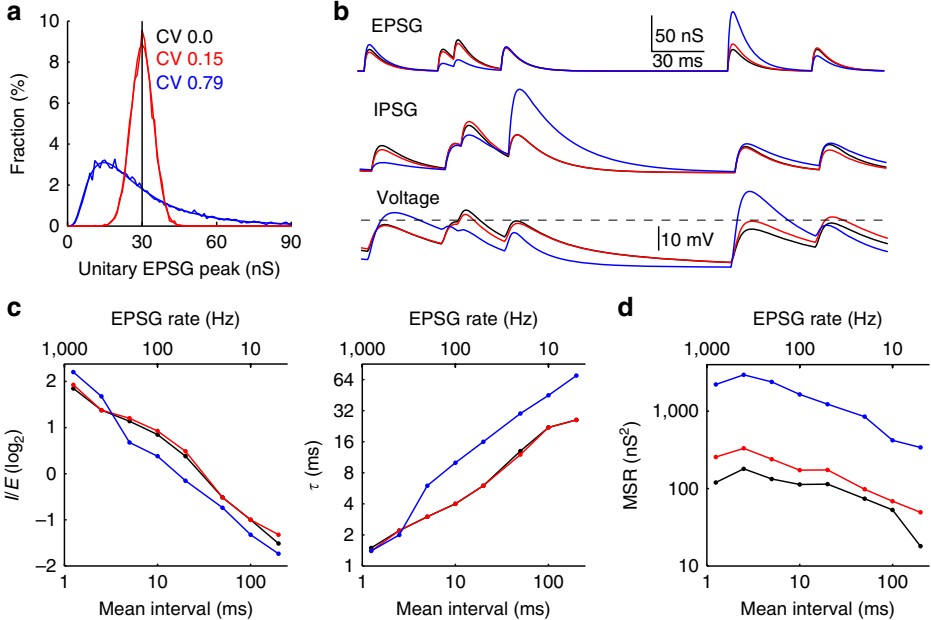

**Figure 6 | Effect of independent variance in unitary EPSG and IPSG amplitudes. (a)** Unitary EPSG had no variance (black, standard condition), or were randomly sampled from a normal distribution with CV of 0.15 (red, $\sigma = 4.5$ nS) or from a log-normal distribution with CV of 0.79 (blue, $\sigma = 23.6$ nS, corresponding to a factor of 2 from the mean). Smooth functions are probability distributions and jagged functions are frequency histograms of 1,000 and 10,000 PSG for normal and log-normal distributions, respectively. The mean EPSG was 30 nS for each distribution, whereas the mean IPSG was determined by the parameter I/E. **(b)** Examples of EPSG, IPSG and voltage (top to bottom) for the three distributions (all simulations utilized the same random sample of PSG intervals). **(c)** Optimal I/E (left) and $\tau$ (right) as a function of EPSG frequency for each amplitude distribution. **(d)** MSR increased with variance.

It is interesting to note that log-normal or similarly skewed distributions of biophysical properties are common[37], and our log-normal condition may be relevant to cortical neurons and others in which synchronous activation of smaller numbers of synapses is presumably more common than larger numbers, resulting in a skewed distribution of EPSG and EPSP amplitudes[38,39].

As another step towards more realistic and complex patterns of EPSG, we considered alternating periods of EPSG at 10 and 100 Hz (Supplementary Fig. 2 and Supplementary Results). Even when 100 Hz occurred only 9% of the time (0.1 s of each 1.1 s period; 18 Hz average, with equal average numbers of EPSG at each frequency), optimal parameters were still more similar to those optimal for 100 than 10 Hz. Thus, brief periods of high frequency have a disproportionately large influence with respect to optimal IPSG, suggesting that our randomized rates may be most comparable to rates achieved in real neurons during periods of mild or moderate stimulus-evoked excitation rather than long-term average excitation (Supplementary Results).

We optimized I/E and $\tau$ with alternative parameters for membrane conductance, E–I delay, and chloride equilibrium potential ($E_{Cl-}$), both with (Fig. 7) and without AP (Supplementary Fig. 1). Although each of these factors had a significant influence, their effects were generally small relative to the effect of EPSG frequency, which still favoured larger and faster IPSG at higher frequencies. Homoeostasis was promoted (MSR was minimized) by addition of AP, low conductance and a brief E–I delay, although these factors mattered less at low EPSG frequencies (Fig. 7, bottom). $E_{Cl-}$ had a smaller influence on MSR, but $-70$ mV was near optimal (Fig. 7, bottom). The influence of each factor is described in detail in Supplementary Results.

**Learning optimal IPSG amplitude and decay time.** From theory one can logically derive anti-Hebbian rules through which a neuron could learn which inhibitory inputs best maintain

homoeostasis given a particular pattern of EPSG[21,40]. Here we show that optimal IPSG can be learned through anti-Hebbian rules (Fig. 8a).

Learning was based only on the local IPSG and the presence or absence of a spike. A spike following IPSG onset was evidence that the inhibitory synapse was too weak, and therefore it was strengthened, whereas absence of a spike had the opposite effect. We tested three learning rules (Methods). Rule 1 was simple and was applied to a neuron with a single inhibitory synapse with $\tau$ pre-specified to be optimal. Rules 2 and 3 were implemented in a neuron with nine inhibitory synapses, each with a distinct decay time (1.5–50 ms) (Fig. 8b). These rules associated spikes with local synaptic 'eligibility', which depended on 'activity' that varied from one synapse to another due to the influence of local $\tau$ on temporal summation (Fig. 8c).

Total inhibition was initially too weak, and all weights increased until approximate E–I balance was attained and nearly half of EPSG caused AP (Fig. 8d). Thereafter, some synapses weakened and others strengthened until IPSG parameters were nearly the same as those that were optimal in minimizing MSR (Fig. 8a). Interestingly, rules 2 and 3 performed better than the simpler rule 1 in minimizing MSR, despite the fact that optimal $\tau$ was assigned with rule 1. When EPSG rate switched from 800 to 5 and back to 800 Hz, near optimal parameters were learned in each case, demonstrating that learning of optimal parameters did not depend on the choice of initial weights (Fig. 8e).

**Spike probability and timing.** As expected, both spikes and spike failures were common with optimal IPSG. With our simple learning rule 1, spike probability (ratio of spikes to EPSG) was almost exactly 1/2 for all EPSG frequencies, whereas for learning rule 2 and minimization of MSR it was less, especially at low frequencies (Fig. 9a). This is explained by asymmetries that are more pronounced at low EPSG frequencies (Fig. 4c,d; Supplementary Fig. 3 and Supplementary Results).

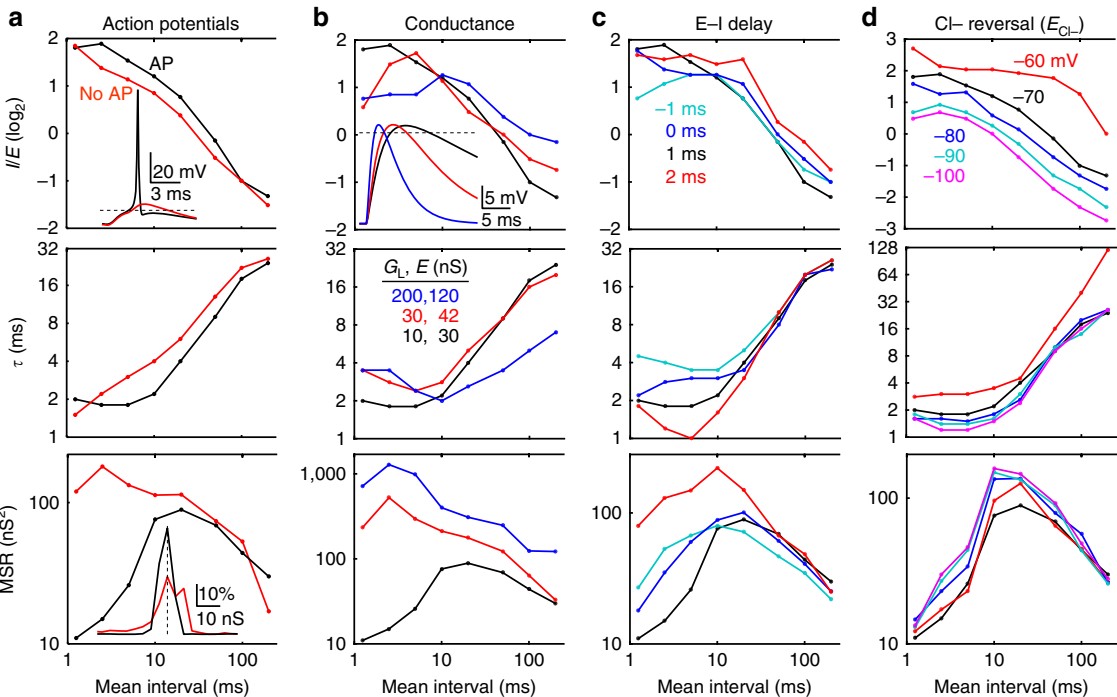

**Figure 7 | Influence of additional factors.** Optimal $I/E$ (top), $\tau$ (middle), and MSR (bottom) as a function of mean EPSG interval. (**a**) Our standard model with (black) and without (red) AP. Data without AP is the same as Fig. 5b,c; data with AP is repeated in **b–d** for comparison. Top inset, sample voltage responses to EPSG at 50 Hz with optimal IPSG. Bottom inset, distribution of residuals at 800 Hz. Dashed lines in insets correspond to $-50$ mV (top) and zero residual (bottom). (**b**) Membrane conductance was low (black; $E = 30$ nS, $G_L = 10$ nS), medium (red; $E = 42$ nS, $G_L = 30$ nS) and high (blue; $E = 120$ nS, $G_L = 200$ nS). The three combinations of $E$ and $G_L$ were chosen so that from resting membrane potential of $-70$ mV, an EPSG would cause an EPSP with peak near $-48$ mV in the absence of IPSG and AP (inset at top). (**c**) E–I delays of $-1$ (cyan), 0 (blue), 1 (black) and 2 ms (red). (**d**) IPSG reversal potentials ($E_{Cl-}$) of $-60$ (red), $-70$ (black), $-80$ (blue), $-90$ (cyan) and $-100$ mV (magenta).

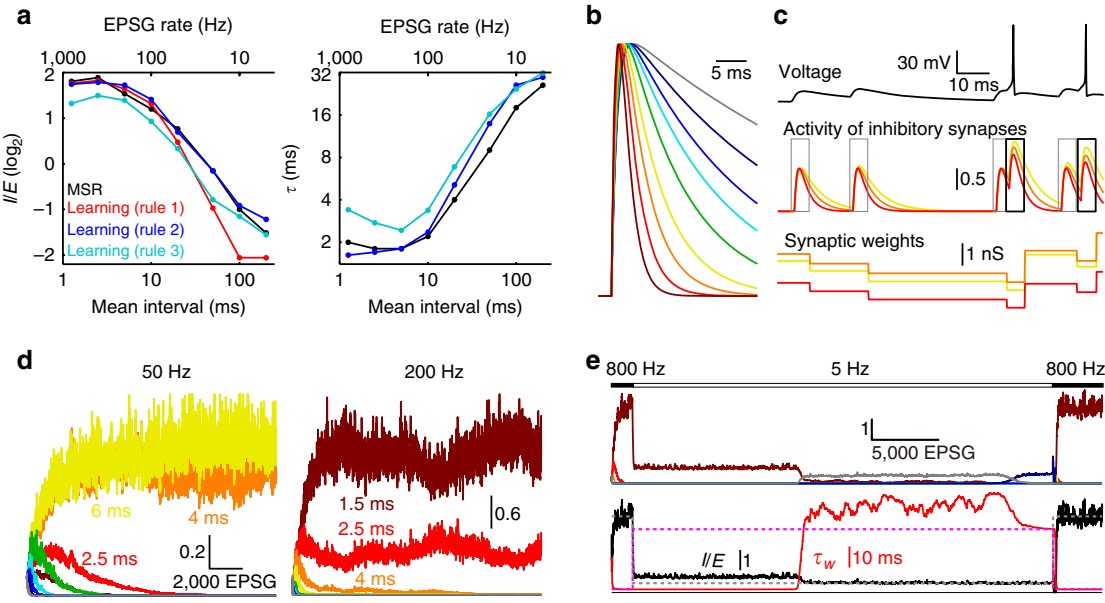

**Figure 8 | Learning optimal IPSG parameters.** (**a**) $I/E$ (left) and $\tau$ (right) as a function of mean EPSG interval for learning rules 1 (red), 2 (blue), 3 (cyan) and for minimization of MSR (black; same as Fig. 5b). With learning, '$\tau$' is a weighted average across the nine synapses (except rule 1). All simulations were performed with AP. (**b**) Normalized IPSG for each of the nine synapses ($\tau$ of 1.5–50 ms). Colour code is maintained in **b–e**. (**c**) Illustration of learning rule 2. From top to bottom, voltage, activity and synaptic weights ($\tau$ of 2.5, 4.0 and 6.0 ms) with EPSG at 50 Hz, selected from a time early in learning, after approximate E–I balance had been attained but before weights were substantially differentiated. Rectangles indicate periods in which activity was averaged to find eligibility; thick black indicates spike occurrence. (**d**) Weight of each synapse ($w_i/E$) as a function of EPSG number at 50 and 200 Hz for rule 2. All weights were initially equal. (**e**) In a separate simulation with rule 2, EPSG frequency was switched from 800 to 5 and then back to 800 Hz. Top, analogous to **d**, except that the learning rate was 2.5 times greater and weights are shown as a moving average with bin size of 50. Bottom, $I/E$ and weighted $\tau$. Dashed horizontal lines indicate $I/E$ (grey) and $\tau$ (magenta) that were learned by rule 2 in **a**.

Although IPSG preserve spike timing[2,41], our criteria for optimality is based only on 'if' and not 'when' a spike occurs. 'If' is more important here, given that our theory and model insures that a spike can only occur within a few milliseconds after onset of an EPSG. Nonetheless, even within this brief window, it is desirable for spike latencies to be short and precise. For $\tau$ of 1–32 ms, all with optimal I/E, each EPSG was always followed by

0 or 1 AP (prior to onset of the next EPSG), and $\tau$ that was optimal in minimizing MSR was nearly optimal in minimizing both the mean and standard deviation of AP latency (Fig. 9b,c). Thus optimization of IPSG with respect to spike occurrence may also optimize spike timing.

**Comparison of theory to experiment.** Figure 10a compares predictions of optimal IPSG from theory with experimental data. Each of the 21 red circles represents the published average $\tau$ from a single type of neuron (Supplementary Table 1), and our estimate of the typical EPSG input rate of that type of neuron under conditions of mild activation (Supplementary Methods). If theory explained all variability in $\tau$, all experimental data would lie on the identity line (Fig. 10b), and the root mean squared error (r.m.s.e.) would correspond to a factor of 1. The actual data deviated from the predictions of theory by a factor of 1.9 for simulations with AP in our standard model (with learning rule 2 or minimization of MSR) (Fig. 8a). Across 20 model neurons (Fig. 7, Supplementary Fig. 1), r.m.s.e. ranged from a factor of 1.8, in the standard model without AP, to 3.1 for the model with the highest conductance and AP.

How accurate is a factor of 2 or 3? A prediction based on the mean of the 21 experimental $\tau$ (9.4 ms) was off by a factor of 3.2, and linear regression by a factor of 1.7. Compared to these, the evidence favoured our model by factors of $10^4$ and $10^{-2}$, respectively (Methods). However, although predictions based on statistics provide a familiar and useful reference, they are not appropriate comparisons. Beyond the fact that statistics do not explain data, our theory and model did not use any knowledge of $\tau$ in deriving predictions (except in the case of learning, where $\tau$ was specified to be between 1 and 50 ms). Since we are unaware of any alternative theory that predicts $\tau$, we compare our results to the 'null hypothesis'. What is the chance that our predictions would be within a factor of 2 given no knowledge of $\tau$? Any $\tau$ is possible, with a leak conductance being the limiting case of 'infinite $\tau$.' As an approximation, we can assume bounds of 1 ms and 1 year, and precision of 1 ms. With no additional knowledge, logic requires that all possible $\tau$ are equally probable. The probability of 'randomly drawing' $\tau$ between 5 and 20 ms (within a factor a 2 from the mean of 10 ms) is $4.8 \times 10^{-10}$. This number is rather arbitrary, but its derivation shows that the match of theory and data is not a coincidence.

Estimation of natural EPSG rates is difficult for a variety of reasons. Although we provide detailed justification for our estimate in each of the 21 types of neuron (Supplementary Methods), uncertainty in these estimates limits confidence in the conclusions drawn from Fig. 10a,b. However, we would expect that the error in our estimates is approximately a factor of 2 (for example, if we estimate 20 Hz, the true rate is probably between 10 and 40 Hz), much less than the factor of 100 that distinguishes neurons with the fastest and slowest EPSG and spike rates (average EPSG rates were 3.7 Hz in locus coeruleus[42], and >500 Hz in medial superior olive in the absence of acoustic stimulation[43]). Therefore uncertainty in our estimates of EPSG rates does not substantially detract from our conclusion that theory can explain variability in IPSG decay times across neurons.

However, we can entirely avoid the difficulty of estimating natural EPSG rates if we instead focus on the important but lesser challenge of explaining only the typical or average IPSG decay time. In other words, why are IPSG decay times on the order of 10 ms across many types of neurons? If we assume that 50 Hz is the median EPSG rate across real neurons, theory predicts optimal $\tau$ to have a median value of 5.1 ms (rule 2 in Fig. 8a), whereas the median across the 21 neurons was 7.0 ms. With this prediction of 5.1 ms, the error (r.m.s.e.) is a factor of 2.8, approximately twice that observed using EPSG rates as an

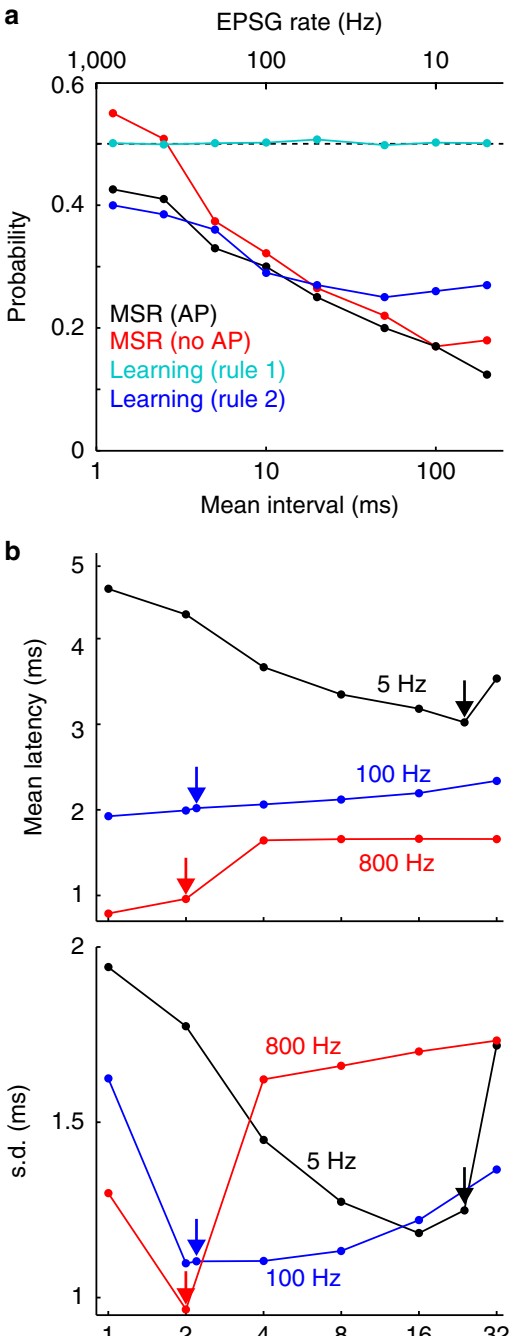

**Figure 9 | Spike probability and timing.** (**a**) Ratio of total spikes to EPSG as a function of mean EPSG interval, with IPSG that minimized MSR in the absence (red) or presence (black) of AP, and after learning according to rule 1 (cyan) and 2 (blue). (**b**) Mean AP latency (top) and s.d. (bottom) from EPSG onset as a function of $\tau$ (with optimal I/E) for EPSG rates of 5 (black), 100 (blue) and 800 Hz (red). Arrows indicate $\tau$ that minimized MSR.

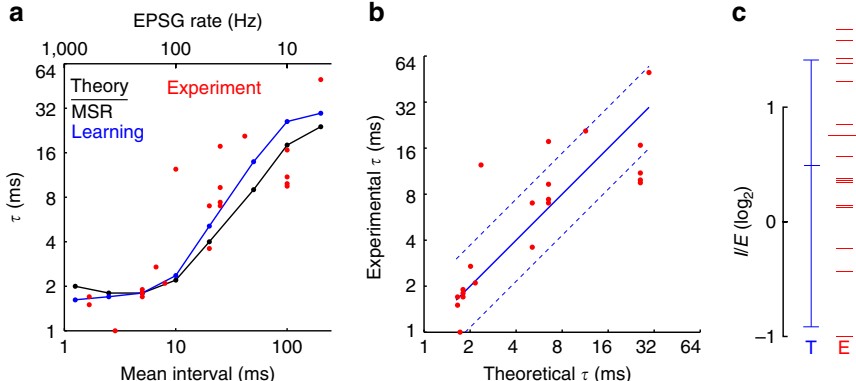

**Figure 10 | Comparison of theoretical optima with experimental data.** (**a**) Experimentally measured average $\tau$ and estimate of typical EPSG rate for 21 types of neurons (red) (Supplementary Table 1 and Supplementary Methods) compared to predictions based on theory (black and blue, same as Fig. 8a). (**b**) Experimental data (circles) and theoretically optimal $\tau$ (identity line). Prediction of theory was based on linear interpolation between data points in **a**. Dashed lines indicate r.m.s.e., which corresponds to a factor of 1.9. (**c**) $I/E$ predicted to be optimal by theory (left; at 10, 40 and 100 Hz, from bottom to top, based on rule 2 in Fig. 8a) and observed experimentally (right; longer line is the mean). Experimental data ($n = 15$) were mostly from cortical pyramidal neurons ($n = 12$), especially primary auditory cortex ($n = 8$), but was distinct with respect to neuronal type, cortical layer, species or laboratory (Supplementary Table 2 and Supplementary Methods).

explanatory factor across neurons (1.9; see above). Therefore the probability of observing this match between theory and experiment doubles, but is only $10^{-9}$. Theory provides a remarkably accurate prediction of typical IPSG decay times, even if we disregard estimates of EPSG rates.

Theory also correctly predicted the mean and range of $I/E$ observed in adult animals *in vivo* (Fig. 10c, Supplementary Table 2). Whereas cortical pyramidal neurons were excluded from measures of $\tau$, they constituted 12 of 15 measures of $I/E$ (Supplementary Tables 1 and 2, Supplementary Methods). We did not attempt to estimate EPSG rates for each of these types of neurons, but we would expect them to be 10–100 Hz, with 40 Hz being our single best guess. For 40 Hz, theory (rule 2; Fig. 8a) predicts optimal $I/E$ to be 1.4, with a range from 10 to 100 Hz of 0.53–2.66 (Fig. 10c, left). The experimental mean was 1.7 with range of 0.5–3.2 (Fig. 10c, right). The 15 measures of $I/E$ deviated from the estimate of 1.4 with a r.m.s.e. corresponding to a factor of 1.7.

Ideally we would like to have estimates of EPSG rates, $\tau$, and $I/E$ in a single type of neuron. We were only able to get such estimates in thalamocortical neurons of LGN and cerebellar granule cells (40 and 50 Hz, 7.0 and 7.4 ms, 1.1 and 1.1, respectively)[31,44,45] (Supplementary Tables and Supplementary Methods). Measurements in both cell types agree with our estimate of optimality (at 50 Hz, 5.1 ms and 1.5).

**Explaining slow IPSG decay in cochlear bushy cells.** In bushy cells of the ventral cochlear nucleus (VCN), $\tau$ is substantially slower than predicted by our standard model (12.4 versus 2.2 ms) (Fig. 10a, Supplementary Table 1). Unlike our standard model, there is little or no phasic IPSG in bushy cells (spike latency is prior to onset of IPSG, 1–2 ms after EPSG onset) as a result of large and fast EPSG and high membrane conductance[41,46–48]. In contrast, phasic IPSG may be present in neighbouring T-stellate cells of VCN, since they have slower EPSG and lower conductance[49]. Likewise they have much faster IPSG decay (1.0 ms) that is close to that predicted by our standard model (Fig. 10a, Supplementary Table 1)[41].

We further examined the same high-conductance model as above ($E = 120$ nS, $G_L = 200$ nS; Fig. 7b) at 100 Hz but with AP. Optimal $\tau$ remained fast (1.8 ms, $I/E = 2.2$), but phasic inhibition remained despite the high conductance (35% of AP occurred after IPSG onset at 1.0 ms). Increasing E–I delay to 2.0 ms eliminated all phasic inhibition and resulted in optimal IPSG that were

slower ($\tau = 9$ ms, $I/E = 1.1$). Next, we made EPSG kinetics $\sim$10-fold faster to match those in bushy cells (which required increasing E to 270 nS to maintain EPSP peak; see Fig. 7b inset), while keeping our standard E–I delay of 1.0 ms. Phasic inhibition was absent and optimal $\tau$ was 34 ms ($I/E = 0.1$). Thus refining our model to better mimic bushy cells resulted in a more accurate prediction of their slow IPSG decay.

## Discussion

Although the field of neurobiology has measured many biophysical properties, it generally lacks an explanation of why those properties are what they are. Why should the IPSG decay time constant be 1.0 ms in T-stellate cells of VCN[41]? Why is it at least eight fold slower in neighbouring bushy cells[41,48]? Although advantages have been found for one set of IPSG parameters over another[19,20,41], we are not aware of any previous predictions of exact optimal values. We used theory and computer simulations to find the IPSG amplitude and decay time that provides the optimal homoeostatic counterbalance to excitation. We further demonstrated that optimal IPSG parameters could be learned through anti-Hebbian rules. Most remarkably, our predictions closely match experimental observations of decay times across 21 types of neuron.

Together with recent experiments[50,51], we have tested a theory of the information contributed to membrane voltage by synapses and ion channels[21,40,52,53]. The theory extends a long line of research relating neuronal function to quantitative principles of information[23,24]. The central idea is illustrated by a balance scale, in which the challenge of measurement is to accurately predict and counterbalance the unknown weight of interest. It was proposed long ago that synaptic inhibition provides the appropriate counterbalance to excitation[23], and there is strong supporting evidence[25]. However, it has not been known exactly what constitutes optimal balance (or 'gain control'). To find this requires a theory that specifies the ideal homoeostatic balance, including the period of time over which that balance should be achieved. Since most theories focus on firing rates or spike patterns (rather than single spikes), optimal E–I balance has usually been defined over periods that are long compared to IPSG decay times[16,17,19,20]. Any particular level of balance over a long time period (for example, a target firing rate over 20 ms) can be achieved with multiple combinations of IPSG decay time and peak amplitude. Thus theories focused on longer time periods are poorly suited for distinguishing IPSG amplitude from decay time,

and we are not aware of any studies of E–I balance that have systematically varied them. In contrast, our theory specifies the optimal E–I balance at the time of a single EPSG and the spike it may cause. Because of this fine timescale, the exact IPSG time course matters, and we can identify a single combination of amplitude and decay time that is optimal (Fig. 3).

The predictions of theory closely matched experimental measures of IPSG amplitude and decay time. One's attention is naturally drawn to the fact that theory correctly predicted the slope of $\tau$ in Fig. 10a, but much more remarkable is that it predicted the actual value of $\tau$ ($y$ offset) within a factor of 2. Indeed, one might mistakenly guess that we 'fit the data' in some way. It is easy to accurately predict data given statistics from the same data, and neuroscientists have had knowledge of IPSG decay times for many years. However, theory should explain and predict data based on principles that are independent of the data. The accuracy of our predictions is remarkable because we did not use any knowledge of $\tau$ in predicting $\tau$. Without knowledge of $\tau$, and in the absence of any formal theory, or intuition, there is no reason to believe that fast decay should be better than slow decay (the extreme limit of slow IPSG decay corresponds to a constant 'leak' conductance). Our prediction was within a factor of 2 despite an infinite range of possible decay times. Although one could question our estimates of natural EPSG rates in the 21 neurons, the error in prediction increased only to a factor of 2.8 when we did not utilize these estimates as a predictive factor, which is still slightly more accurate than the factor of 3.2 for a prediction based on the mean decay time across the 21 neurons. Thus, if we disregard our estimates of natural EPSG rates (Fig. 10a,b), then we cannot claim to have explained variability in IPSG decay across neurons, but we can still claim to have explained the typical or average IPSG decay time.

Theory was also able to predict the mean and range of experimentally measured ratios of peak synaptic inhibition to excitation (Fig. 10c). We had data on both I/E and $\tau$ for only 2 of the 21 neurons (cerebellar granule cells and thalamocortical neurons), and theory accurately predicted the combination of parameter values in those neurons (see Results).

As a test of theory, the present results apply more to neurons that receive few rather than many strong excitatory synapses. For a variety of reasons we did not include cortical pyramidal neurons among the 21 types of neurons from which we obtained IPSG decay times (Supplementary Methods). The greatest difficulty is that the presence of a large number of excitatory synapses make it possible to generate many patterns of synaptic excitation, and yet we have little knowledge of the actual patterns. The knowledge we do have indicates that synchronous but temporally sparse excitation causes large and discrete EPSPs, at least in some cortical pyramidal neurons[38,39]. Although our standard model had large and discrete EPSPs, it lacked the variability that would be expected in a neuron with many excitatory synapses. However, limited results from an alternative model, having the high variance and skewed distribution of EPSG amplitudes that might be expected in a cortical pyramidal neuron, led to qualitatively similar conclusions (Fig. 6).

Despite our confidence in the theory, we were surprised by how well it predicted the experimental data given our neuronal model. The theory specifies that optimality should depend on numerous factors that contribute to membrane excitability, yet our model neuron was simple and generic, with a single electrical compartment and no active intrinsic properties (or only AP). Our intention was to mimic a typical neuron, and in principle such a generic 'average model' may be the best single model for explaining variance in IPSG across diverse neurons. However, a more detailed model should produce a more accurate prediction with respect to a particular neuron. The largest error of our standard model occurred with bushy cells of VCN, in which $\tau$ is more than five fold slower than predicted (Fig. 10a,b)[41,48]. When we eliminated phasic inhibition to better mimic bushy cells, optimal $\tau$ was slower and more similar to that found in bushy cells. Thus the relatively small errors in prediction based on our simple model may be reduced further through detailed models of specific types of neuron.

Evidence for the theory would be further strengthened if it can correctly predict additional parameters of inhibition, and especially combinations of multiple parameters. For example, multiple specializations speed conduction through inhibitory pathways so that E–I delay is brief or even negative[3,11,14,31,49,54], consistent with our results (Fig. 7c, bottom). In layer 6 of auditory cortex, one neuronal population has negative E–I delay and large I/E ($-1.6$ ms and 3.0), whereas another population has the opposite combination ($+1.6$ ms and 0.6)[11]. Neurons in the medial superior olive are of particular interest because they are exceptionally fast. They receive EPSG at over 500 Hz (ref. 43), have fast IPSG decay (1.7 ms), negative E–I delay ($-0.4$ ms), and unusually negative $E_{Cl-}$ ($-90$ mV)[14,55–57]. At the opposite end of the spectrum are neurons of the inferior olive, which fire at about 2 Hz (ref. 58) and have exceptionally slow IPSG decay[59].

A combination of physiology and simulations could be used to test the theory's ability to predict multiple IPSG parameters over the course of development, or in response to direct manipulation of EPSG patterns (for example, at the retinotectal synapse[13]). Membrane properties become faster over development, including the rate of IPSG decay[56]. The present results suggest that faster IPSG decay could result from anti-Hebbian learning in response to an increasing prevalence of brief EPSG intervals over the course of development (whether or not average rates increase).

Real neurons have numerous conductances that were lacking in our models, and we have proposed that many of these work to maintain homoeostatic balance as described here for IPSG[21,40]. Like phasic IPSG, some low-threshold K+ channels activate during the rising phase of EPSP (for example, Kv1) and complement IPSG in keeping EPSP peak close to spike threshold[14]. The theory should predict the properties of voltage-gated ion channels as it did IPSG. For example, the theory correctly predicted the experimental observation that T-type calcium channels are homoeostatic in counterbalancing opponent IPSG in thalamocortical neurons of LGN[50]. The more general theory proposes differences between sensory and motor neurons, and it correctly predicted evidence that T-type calcium channels are not homoeostatic but cause bursts of spikes in motor thalamus[51]. In addition to predicting the properties of a type of neuron, theory may also be useful in explaining variability in ion channel parameters across neurons of the same type, particularly by indicating which combinations of ion channel parameters work in a concerted manner to maintain near optimal homoeostatic excitability[60,61].

Anti-Hebbian rules have previously been shown to optimize the amplitude of synaptic inhibition to maintain sensitivity of spikes to synaptic excitation and sensory stimuli[25,62,63]. Here we have shown for the first time that anti-Hebbian rules could enable simultaneous learning of optimal IPSG decay times as well as amplitudes, as predicted by theory[21,40] (Fig. 8). Thus adjustment of IPSG parameters could restore homoeostatic excitability within minutes following a change in the temporal pattern of EPSG[25]. By providing a plausible mechanism for dynamic optimization, our results with learning strengthen our conclusion that IPSG are optimized for homoeostatic balance in accord with theory.

## Methods

**Model neuron.** We simulated a single-compartment neuron using NEURON software (version 7.3)[64]. The model neuron had a membrane of 24,058 $\mu m^2$ and

capacitance ($C_m$) of 0.24 nF (1 µF cm$^{-2}$), as previously described[65]. It included a leak conductance ($G_L$), EPSG ($g_{Exc}$) and IPSG ($g_{Inh}$). Simulations with AP also included a Hodgkin–Huxley-type, voltage-dependent sodium conductance ($g_{Na}$) and potassium conductance ($g_K$). In simulations with AP, the dynamics of the membrane potential ($V$) were governed by equation (1).

$$C_m \frac{dV}{dt} = g_{Na}m^3h(E_{Na} - V) + g_K n^4(E_K - V) + g_{Leak}(E_{Leak} - V)$$
$$+ g_{Exc}(t)(E_{Exc} - V) + g_{Inh}(t)(E_{Inh} - V) \tag{1}$$

Reversal potentials ($E$) for each conductance were 50, $-90$, $-70$, 0 and $-70$ mV, respectively. In some simulations, the IPSG reversal potential ($E_{Inh}$) had other values (Fig. 7d). The time unit for simulations was 0.1 ms in simulations with AP, and 0.25 ms without AP.

**Synaptic conductances.** Post-synaptic conductances (PSG) were intended to mimic those mediated by AMPA-type glutamate receptors and GABA$_A$/glycine receptors. They followed the form of equation (2).

$$PSG = e^{-t/\tau_{decay}} - e^{-t/\tau_{rise}} \tag{2}$$

where $t = 0$ at the onset of the conductance, $\tau_{rise}$ is the rise time constant, and $\tau_{decay}$ is the decay time constant. For EPSG, these were 0.45 and 3.0 ms, respectively. For IPSG, $\tau_{rise}$ was 0.9 ms, whereas IPSG decay time ($\tau$) was varied between 1 and 160 ms. In our standard model, IPSG onset always followed EPSG onset by 1.0 ms.

**Action potentials.** We designed AP to be simple and brief. By making AP brief we were able to measure residuals for every EPSG. Another reason for using simple and brief AP is that AP in real neurons activate a diversity of conductances, which typically cause an after-hyperpolarization lasting at least a few milliseconds. We expect that these conductances are homoeostatic and work synergistically with IPSG[21,37]. Although it is of interest to investigate how these conductances interact with IPSG, we avoided this issue by having a single K$+$ conductance with rapid decay.

We modified the parameters of a previous model (available from 'NEURON ModelDB')[65]. The absolute refractory period was $<1.0$ ms, which was the minimum interval between EPSG and enabled us to measure a residual at the time of each EPSG (every EPSG in our simulations at 800 Hz was able to evoke an AP if it was sufficiently large). AP duration at half of peak amplitude was 0.2 ms, and $G_{K+}$ at 0.5 ms after AP peak was 6 nS, 0.1% of its maximum and 60% of our standard $G_L$.

AP thresholds were adjusted to be near $-50$ mV, which was designated as 'spike threshold' in simulations without AP. The mean maximal sub-threshold EPSP (measured in response to 'threshold ESPG' as part of our method of finding residuals) was $-52.3$, $-50.5$ and $-50.0$ mV with EPSG at 5, 100 and 800 Hz, respectively, in simulations with optimal IPSG.

**Ensemble of EPSG.** In our standard model, each EPSG was of identical unitary amplitude ($E = 30$ nS). EPSG amplitude was larger in some simulations (Fig. 7b) and variable in others (Fig. 6). The most critical variable was the mean interval between EPSG. Except in simulations in which an 'ensemble' consisted of only 2 EPSG (Fig. 2), EPSG ensembles were generated by randomly sampling from geometric interval distributions (the discrete analogue of an exponential distribution) with a discrete unit of 1.0 ms. Thus an EPSG interval could be 1.0, 2.0, 3.0 ms and so on. Mean EPSG frequencies varied from 1 to 800 Hz (mean intervals of 1,000 to 1.25 ms). Although EPSG intervals were randomly sampled at each frequency, sampling was only performed once for each frequency. Thus the same sequence of intervals was used for every simulation of a given frequency (Figs 3a and 6b).

MSR was found with ensembles of 1,000 EPSG for each combination of parameters and at each frequency, and for each neuronal model. However, 5,000 EPSG were used in the case of our standard model at 5 Hz. Testing with 4,000 additional EPSG did not result in any change to optimal parameter values relative to 1,000 EPSG, but slightly reduced MSR (18.6–16.6 nS$^2$). With log-normal variance in unitary PSG, we used 10,000 to sufficiently sample the larger space of both amplitudes and intervals. The number of EPSG tested with learning was chosen to reach stable synaptic weights (Fig. 8d) (see below).

**Residuals and MSR.** At the time of each EPSG, we measured 'distance from optimality' as previously described[21]. We refer to this distance as a 'residual.' After finding the 'real' voltage in response to an EPSG ensemble, we performed additional test simulations to find how much larger or smaller each EPSG would need to have been in order for the EPSP peak to reach exactly to spike threshold (Fig. 2a). Critically, the $n$th residual depended on membrane properties at the time of EPSG$_n$, including IPSG$_n$, but it did not depend on EPSG$_{n+1}$ and other future events (Fig. 2a). Therefore, to find the $n$th residual, the voltage and conductance up to the $n$th synaptic event was kept for the test simulation, but later EPSG and IPSG were discarded. Test EPSG were injected with onset at the time of the real EPSG$_n$, making it larger or small as needed so that the peak of the test EPSP was as near as possible to spike threshold (AP threshold, or $-50$ mV in simulations

without AP). The increment of test EPSG was 0.3 nS. This increment determined the precision with which we estimated the threshold EPSG and residual. In simulations without AP, the peak of the 'threshold EPSP' was $50 \pm 0.01$ mV (mean $\pm$ s.d.).

The residual was defined as the negative of the difference between the 'threshold EPSG' and the 'real EPSG'. For display purposes only, we used the negative of this difference so that positive residuals corresponded to EPSP peaks that were positive of $-50$ mV (Figs 3a and 4a,d, Supplementary Fig. 3c). For a given ensemble of EPSG, the optimal IPSG parameters were taken to be those that minimized the MSR.

In simulations without AP, occasionally there were cases in which membrane voltage was above 'spike threshold' of $-50$ mV at the time of EPSG onset. This almost never occurred at low EPSG frequencies, but it occurred with 18.7% of EPSG at 800 Hz in our standard model with optimal IPSG parameters. In such cases, the threshold EPSG should bring the EPSP to $-50$ mV, but the EPSP had no peak in this case and the voltage trajectory continued downwards. This creates a problem for measuring the residual. To measure the residual, we defined the 'threshold EPSG' to be the EPSG that caused the mean voltage at 1–3 ms after EPSG onset to be nearest to $-50$ mV.

To find optimal parameter values, $\tau$ was tested across a range of 1–50 ms, but up to 120 ms in Fig. 3d, and 160 ms with $E_{Cl-}$ of $-60$ mV (Fig. 7d). For values of $\tau$ that were near optimal in minimizing MSR, we tested increments of 0.1 ms at 1–2 ms, 0.2 ms at 2–3 ms, 0.5 ms at 3–5 ms, 1.0 ms at 5–16 ms, 2.0 ms at 16–50 ms and 10.0 ms at 50–160 ms, although less in some instances. Values of $I/E$ that were near optimal for a given $\tau$ were tested in increments of 0.1 ($\Delta I = 3$ nS) for $\tau$ of 1–22 ms, and 0.05 for larger $\tau$. However, for 5, 100 and 800 Hz with our standard model (Fig. 3d), increments were 0.01 for $\tau$ above 14 ms.

**Learning.** Since our goal was only to demonstrate the feasibility of anti-Hebbian learning, we made little effort to optimize the learning rules, or to make them realistic in all respects. For simplicity, learning occurred following each synaptic event, rather than in real time.

We tested three rules. Rule 1 was simple and was applied to a neuron with a single inhibitory synapse. It lacked a dynamic 'presynaptic' term and thus was not 'associative' in a meaningful sense. IPSG decay time was pre-specified to be optimal for the EPSG frequency, and thus the only objective was to learn the optimal homoeostatic value of $I/E$. Learning with rule 1 occurred following each synaptic event ($n$) according to equation (3).

$$w_{n+1} = w_n + \alpha v_n \tag{3}$$

The learning rate $\alpha$ was 0.6 nS per synaptic event. The weight of the inhibitory synapse ($w$) increased or decreased depending on whether an AP did ($v = 1$) or did not occur ($v = -1$) during the 'spike period,' which was $-0.5$ to 4.5 ms from IPSG onset, or prior to onset of the next IPSG if the next IPSG occurred within $<4.5$ ms. The synaptic weight was updated at the end of the spike period, and thus $w_n$ was effective from 4.5 ms after IPSG$_n$ to 4.5 ms after IPSG$_{n+1}$ (Fig. 8c).

Rules 2 and 3 addressed the greater challenge of learning IPSG decay time as well as amplitude. The model neuron had nine inhibitory synapses, each having synchronous activation 1.0 ms after each EPSG, but with a distinct decay time ($\tau = 1.5$–50 ms; Fig. 8b). The IPSG at synapse '$i$' and time '$t$' depended on synaptic weight ($w_{i,t}$) and activity ($u_{i,t}$) (equation (4)).

$$IPSG_{i,t} = w_{i,t}u_{i,t} \tag{4}$$

'Activity' was analogous to 'presynaptic activity' in conventional associative rules, and corresponds to the time course of GABA$_A$ or glycine receptor activation (unitary activity at each synapse had a peak of 1), whereas the 'weight' can be understood as the number of receptors at the synapse. The IPSG is decomposed into 'weight' and 'activity' so that the regulation of weight (learning, or induction of plasticity) can be independent of weight (memory, or expression of plasticity). (This could occur if there are GABA$_A$ receptors that regulate learning but do not allow current to pass.)

Learning at the time of a synaptic event should depend on the information available about the causal relation between inhibitory activity and spike generation. The 'eligibility' for learning ($\bar{u}_{i,n}$) was the average synaptic activity during the period $-1$ to 4.0 ms from IPSG onset (0–5 ms from EPSG onset), or until the onset of the subsequent IPSG if it occurred in $<4.0$ ms (indicated by rectangles in Fig. 8c). The spike period during each synaptic event was also 5.0 ms in duration, but was delayed by 0.5 ms from the eligibility period. Although this was a minor detail, the principle was to account for the causal delay expected between inhibitory activity and spike generation. Defining eligibility and spikes in these rather long periods was simplistic, and as a consequence, learning sometimes depended in part on inhibitory activity that occurred just after a spike. This would not occur in a more realistic model, which could have used an exponentially decaying eligibility trace.

Weights were updated at the end of the spike period according to rule 2 (equation (5)) or 3 (equation (6)).

$$w_{i,n+1} = (2^{v_n} - (2^{v_n} - 1)e^{-\alpha\bar{u}_{i,n}})w_{i,n} \tag{5}$$

$$w_{i,n+1} = w_{i,n} + \alpha\bar{u}_{i,n}v_n/\tau_i \tag{6}$$

The learning rate α was 60 and 0.9 nS per synaptic event in learning rules 2 and 3, respectively. Learning with rule 2 was faster at stronger synapses (as measured in nS), but had no direct dependence on decay time. In contrast, learning with rule 3 had no dependence on synaptic weight, but was inversely proportional to τ and was thus slower at synapses with slower decay.

Simulations of learning were run until synaptic weights stabilized. We used 2,000 synaptic events for rule 1, 15,000 with rules 2 and 3 (Fig. 8d), and 35,000 synaptic events for 'reversal learning' (Fig. 8e). The 'learned parameter values' (Fig. 8a) were found by averaging across the last 100 synaptic events. For comparison to our standard parameters '$I/E$' and τ (Fig. 8a), the total phasic peak IPSG amplitude '$I$' across all synapses was the sum of the weights ($w_i$) across the nine synapses, and 'weighted τ' was the weighted sum of the decay time constants across the nine synapses, where the weights were the synaptic weights ($w_i$).

**Statistical analysis.** Although it is not a fair comparison (see Results and Discussion), we compared the evidence for our model of τ to two other models, one based on the mean τ across the 21 types of neurons, and the other based on linear regression of the 21 τ on EPSG frequency. The 'odds' are defined as the ratio of the probability of the data given one model to the probability of the data given the other model, which corresponds to the evidence favoring one model over another[66]. We assumed that errors were normally distributed, and that the parameter σ was equivalent to the r.m.s.e. calculated from the model and sample data. Since it follows that the errors were conditionally independent, the probability of the 21 data points was the product of the 21 probabilities. The probability of the data was $10^4$ times greater given the predictions of our model relative the prediction based on the mean τ, and thus the evidence for our model was $10^4$ times greater. The evidence favoured linear regression over our model by a factor of ~100.

**Data availability.** Code and additional parameter values used in the NEURON model are available from the authors upon request.

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

## Acknowledgements

We thank Larry Trussell, Gary Westbrook, Mike Wehr and Gerard Borst for helpful discussions. Research was supported by a grant to C.D.F. from the National Research Foundation of Korea (2015-R1A2A2A01004185).

## Author contributions

J.K.K. performed simulations, analysed data, and made figures. C.D.F. and J.K.K. designed simulations, collected published data from the literature, and wrote the manuscript.

## Additional information

**Competing financial interests:** The authors declare no competing financial interests.

