## [Peer Review File · Nature Communications]

Reviewers' Comments:

Reviewer #1 (Remarks to the Author):

The authors show that the hypothesis of “predictive homeostasis” is able to quantitatively predict, with surprising accuracy, the decay times of inhibitory conductances at a variety of neuron types. The authors also show, in a proof-of-concept way, as anti-Hebbian rules could learn the proper decay time by making use only of local (in space and time) information.

I think that the results presented in this manuscript are novel, convincing, intriguing and doubtless important. The manuscript is clear, well written and, more personally, I have enjoyed reading it. I do not hesitate in recommending publication.

Minor:

Symbols used in the main text do not match the corresponding symbols in the methods (e.g. G_L in the main text, and g_{leak} in the methods).

Please check the ms for typos.

Reviewer #2 (Remarks to the Author):

In the present article, Kim and Fiorillo use computational modeling to determine whether optimal biophysical parameters of synaptic inhibition can be theoretically predicted depending of the rate of excitatory events received by a neuron. Optimality is defined as providing the best homeostatic balance between excitation and inhibition, on a short time scale, in response to unitary excitatory events. By testing a wide range of values for the amplitude and kinetics of inhibition on model neurons receiving 1-800 Hz of synaptic excitation, the authors demonstrate that optimal inhibition increases in amplitude and decay rate as synaptic excitation frequency increases. In order to prove the relevance of this theoretical prediction, the authors then confront their results with experimental measurements of the properties of synaptic excitation and synaptic inhibition in 21 different neuronal types from the mammalian brain. They show that the optimal time constants of decay of inhibition provided by their model accurately predict the time constants measured electrophysiologically in the neuronal types considered. In general, the

manuscript is very well written, the theory proposed by the authors is really interesting and attempts to answer a really important question, i.e. to determine whether properties of inhibitory and excitatory inputs are co-tuned in order to ensure optimal homeostasis and information transfer. Nevertheless, I have some concerns about the main conclusions of the study, and in particular how the theoretical predictions are linked to experimental data.

MAIN CONCERNS

1) The authors state at the very beginning of the manuscript that they will only consider the case of excitatory synapses where single EPSCs are able to trigger action potentials (very large EPSCs). From a theoretical and modeling perspective, this choice is understandable, but when considering the mammalian brain, this case might only apply to a minority of cell types, since most cortical neurons will receive thousands of excitatory synapses, and will fire action potentials only in response to (more or less) synchronous release of glutamate by tens or hundreds of presynaptic neurons. And indeed, the authors chose to discard cortical cells from their analysis for this reason, although these neurons have probably been most thoroughly characterized from a biophysical point of view (including their synaptic properties). This would not be a problem if the authors did not draw slightly overreaching conclusions from their study. From reading the conclusion, we may feel that the authors have found a general rule that explains the properties of phasic synaptic inhibition in any neuronal type of the mammalian brain, although it has to be understood as explaining the properties of phasic inhibition in the restricted case where neurons receive (very) strong excitatory inputs. The authors should make this point clearer in the discussion and specify what would be the consequences of their theory on IPSP's shape in the case of a cell receiving many small EPSPs (sensory cortical cells). Moreover, it is unclear whether all 21 neuronal types chosen by the authors receive that type of excitatory inputs.

2) Related to the first point, the choice of the 21 neuronal types considered (Suppl. Table 1 and Figure 8) and of their respective EPSP rates does not seem as straightforward as the authors may claim it. In my opinion, this is the most critical point of the current study, because the experimental validation of the theory totally relies on the EPSP rates affected to the different neuronal types where the biophysical properties of synaptic inhibition have been experimentally determined. Some values and choices are rather confusing. Based on a previous study (Fiorillo et al., 2014), the authors postulate that, in general (independent of neuronal types), the probability of firing of neurons in response to EPSPs can be assumed to be 50%. This is only partly true if we consider that i) much variability is present in each neuronal type, and ii) this probability is 100% in some cells and only 25% in others. The experimental results arguing in favor of the 50% postulate have been mainly obtained from sensory nuclei (LGN and MNTB) (Fiorillo et al., 2014). Moreover, the Figure 7a shows that the predictive homeostasis theory gives a firing rate lower than 50% (between 20 and 40% for increasing rates of EPSPs) in most of the cases studied (except for learning rule 1 where IPSP decay rate is already set to its optimal value before

learning). The authors should explain this discrepancy. The arguable choice of 50% for the relationship between spike rate and EPSG rate is critical, because for a number of the 21 neuronal types used in the current study, the authors extrapolate EPSG rates based only on the firing frequency of neurons (if a neuron fires at 10Hz, then we can assume that EPSG rate is the double, 20Hz). As said before, in a cell type where the probability of an EPSC to trigger a spike is 100% (or 25%), this extrapolation cannot be used. Moreover, if we take the specific example of monoaminergic and basal ganglia neurons (top of supplementary table 1), it is known that most of these neurons are spontaneously active in the absence of excitatory inputs (in slices for instance, case of SNc DA neurons, SNr GABAergic neurons, DR 5HT neurons, striatum ACh neurons). A large part of the firing activity of these neurons is due to their intrinsic properties and not to incoming glutamatergic inputs. To my knowledge, the 50% relationship between EPSG rate and firing frequency in these neurons is purely extrapolated from neuronal types that are mainly activated by synaptic inputs (such as thalamic sensory neurons or auditory nucleus neurons). The monoaminergic and basal ganglia neurons are particularly critical for the authors' conclusions, as they display some of the slowest EPSG rate (10Hz) of the cohort of neuronal types used for Figure 8. This class of neurons is just one example, but in general, the calculation/definition (and often extrapolation from sparse experimental data) of EPSG rates in the different neuronal types selected for the current study is rather vague and debatable (3 pages of the supplementary materials are necessary to justify these values). This point constitutes my main criticism of the article, as Figure 8 represents the strongest argument in favor of the theory proposed by the authors. If most of the values of EPSG rates are questionable, then the whole theory collapses or is at least strongly undermined.

Reviewer #3 (Remarks to the Author):

This manuscript explores the potential role of synaptic inhibition in counter-balancing synaptic excitation in what the authors describe as a "homeostatic balance" between excitation and inhibition. The computational studies appear well motivated, executed, and described. That said, I do not fully agree with the interpretation (and for that matter, the title of the manuscript) as it refers to homeostasis.

However, I am not insisting on major terminology changes, but I just want to reiterate that "homeostasis", in my mind, is not what this paper is about.

For other (mostly minor) comments, please refer to the attached Word document with tracked comments and suggested edits.

We are pleased that all three reviewers found our manuscript to be interesting and important.

We have revised the manuscript to better clarify the caveats and limitations of our results. We have added new results with action potentials and variable EPSP amplitudes. We performed these during the review process, but some of this new data addresses a concern of reviewer 2. The new data resulted in one new figure. We also split a large figure into two, and thus the total number of figures in the main manuscript has increased from 8 to 10.

Reviewer's text is blue, response is black.

Reviewer #1 (Remarks to the Author):

The authors show that the hypothesis of "predictive homeostasis" is able to quantitatively predict, with surprising accuracy, the decay times of inhibitory conductances at a variety of neuron types. The authors also show, in a proof-of-concept way, as anti-Hebbian rules could learn the proper decay time by making use only of local (in space and time) information.

I think that the results presented in this manuscript are novel, convincing, intriguing and doubtless important. The manuscript is clear, well written and, more personally, I have enjoyed reading it. I do not hesitate in recommending publication.

Minor:

Symbols used in the main text do not match the corresponding symbols in the methods (e.g. G_L in the main text, and g_{leak} in the methods). Please check the ms for typos.

We have checked for typos and corrected the notation.

Reviewer #2 (Remarks to the Author):

In the present article, Kim and Fiorillo use computational modeling to determine whether optimal biophysical parameters of synaptic inhibition can be theoretically predicted depending of the rate of excitatory events received by a neuron. Optimality is defined as providing the best

homeostatic balance between excitation and inhibition, on a short time scale, in response to unitary excitatory events. By testing a wide range of values for the amplitude and kinetics of inhibition on model neurons receiving 1-800 Hz of synaptic excitation, the authors demonstrate that optimal inhibition increases in amplitude and decay rate as synaptic excitation frequency increases. In order to prove the relevance of this theoretical prediction, the authors then confront their results with experimental measurements of the properties of synaptic excitation and synaptic inhibition in 21 different neuronal types from the mammalian brain. They show that the optimal time constants of decay of inhibition provided by their model accurately predict the time constants measured electrophysiologically in the neuronal types considered. In general, the manuscript is very well written, the theory proposed by the authors is really interesting and attempts to answer a really important question, i.e. to determine whether properties of inhibitory and excitatory inputs are co-tuned in order to ensure optimal homeostasis and information transfer. Nevertheless, I have some concerns about the main conclusions of the study, and in particular how the theoretical predictions are linked to experimental data.

The reviewer has raised two critical issues. We had anticipated these same issues, and had addressed both, although partly in Supplementary Materials. Now we have added new data and addressed these issues in the main manuscript. We performed new simulations to address the first issue, after the original submission but prior to receiving the reviews.

MAIN CONCERNS

1) The authors state at the very beginning of the manuscript that they will only consider the case of excitatory synapses where single EPSCs are able to trigger action potentials (very large EPSCs). From a theoretical and modeling perspective, this choice is understandable, but when considering the mammalian brain, this case might only apply to a minority of cell types, since most cortical neurons will receive thousands of excitatory synapses, and will fire action potentials only in response to (more or less) synchronous release of glutamate by tens or hundreds of presynaptic neurons.

This issue was discussed in Supplementary Materials, since it is relevant to estimating EPSC rate. There we point out that it is only the strong “class 1 drivers” on proximal dendrites that are important in causing spikes (to use

the terminology of Murray Sherman). Only those synapses are relevant to our “EPSP rate.” Although the total number of excitatory synapses (and presynaptic neurons) may be 1000 or 10,000 for a pyramidal neuron, the number of strong synapses that cause spikes is probably closer to 100 or less. In LGN, a thalamocortical neuron receives only 1 to 5 retinal inputs on its proximal dendrites, and these have been shown to cause every spike. But it has about 1000 weak corticothalamic synapses on its distal dendrites. A paper came out a few years ago showing that the number of coactive feedforward excitatory synapses in neurons in V1 is about 10 to 30 (in response to visual stimulation), if I remember correctly. I do not recall the authors now.

And indeed, the authors chose to discard cortical cells from their analysis for this reason, although these neurons have probably been most thoroughly characterized from a biophysical point of view (including their synaptic properties).

Yes, difficulty in estimating natural EPSP rates was the primary reason, due to large numbers of synapses, sparse activity, and the potential influence of dendritic filtering. Other concerns were the multiple sources of inhibitory input, space-clamp problems with respect to measuring IPSP decay, and the complexities and burden of making a multi-compartment model for our simulations.

This would not be a problem if the authors did not draw slightly overreaching conclusions from their study. From reading the conclusion, we may feel that the authors have found a general rule that explains the properties of phasic synaptic inhibition in any neuronal type of the mammalian brain, although it has to be understood as explaining the properties of phasic inhibition in the restricted case where neurons receive (very) strong excitatory inputs. The authors should make this point clearer in the discussion and specify what would be the consequences of their theory on IPSP's shape in the case of a cell receiving many small EPSPs (sensory cortical cells).

We now point out this limitation explicitly in the revised Discussion. In addition, to consider the case of a neuron with many excitatory synapses, we have performed new simulations with the sort of skewed, high variance distribution of EPSP amplitude that we would expect in a neuron with many excitatory synapses (Fig. 6). Although optimal IPSP parameters were

altered (Fig. 6), the results were consistent with our general conclusions. This is described in the revised Results. Below is the relevant revised text in Results and Discussion.

Results:

Our standard model was simplified in using unitary EPSC and IPSC that were of constant amplitude over time. However, variability *in vivo* is much less than one would expect given typical conditions in brain slices³⁵. Under naturalistic conditions at the calyx of Held, EPSC at 40 Hz had a coefficient of variation (CV) of 8%³⁶. We performed additional simulations with unitary synaptic conductance amplitudes having the same mean (30 nS) but CVs of 15% and 79%, with normal and log-normal distributions, respectively (**Fig. 6a,b**) (in the latter case, the standard deviation corresponded to 15 to 60 nS, a factor of 2 from the mean). EPSC and IPSC varied independently (although we would expect moderate covariance under natural conditions). Relative to our standard model, the lesser variance had virtually no effect on optimal IPSC parameters, whereas larger variance with a skewed distribution decreased the amplitude and more than doubled the decay time at all but the highest frequencies (**Fig. 6c**). MSR increased with variance, as expected (**Fig. 6d**). It is interesting to note that log-normal or similarly skewed distributions of biophysical properties are common³⁷, and our log-normal condition may be relevant to cortical neurons and others in which synchronous activation of smaller numbers of synapses is presumably more common than larger numbers, resulting in a skewed distribution of EPSC and EPSP amplitudes^{38,39}.

Discussion:

As a test of theory, the present results apply more to neurons that receive few rather than many strong excitatory synapses. For a variety of reasons we did not include cortical pyramidal neurons among the 21 types of neurons from which we obtained IPSC decay times (see Supplementary Methods). The greatest difficulty is that the presence of a large number of excitatory synapses make it possible to generate many patterns of synaptic excitation, and yet we have little knowledge of the actual patterns. The knowledge we do have indicates that synchronous but temporally sparse excitation causes large and discrete EPSPs, at least in some cortical pyramidal neurons^{38,39}. Although our standard model had large and discrete EPSPs, it lacked the variability that would be expected in a neuron with many excitatory synapses. However, limited results from an alternative model, having the high variance and skewed distribution of EPSC amplitudes that might be expected in a cortical pyramidal neuron, led to qualitatively similar conclusions (Fig. 6).

In a neuron with many excitatory inputs, EPSP could be occurring constantly, resulting in a Gaussian-like distribution of EPSP amplitudes and membrane voltages, and a “random walk” over time. I am not aware of any evidence that any neuron actually does exhibit such a pattern. We did not examine this specific possibility in our simulations, nor did we mention it as a possibility in the text. We did cite DeWeese and Zador (2006) and Softky and Koch (1993) (references 38 and 39), who addressed the possibility of a ‘random walk,’ and provided evidence that at least some cortical pyramidal neurons do not have such a pattern, but instead exhibit highly skewed distributions of EPSP and EPSP amplitudes due to excitation is that highly synchronous across synapses, and sparse over time.

Moreover, it is unclear whether all 21 neuronal types chosen by the authors receive that type of excitatory inputs.

In Supplementary Methods we provided all of the directly relevant information we have on each of the 21 types of neurons. Certainly some of our selected neurons receive many excitatory inputs. We included Purkinje neurons despite this concern (line 331 of Supplementary Methods), in part because their spike rate is well known and quite regular, and this gave us more confidence in our estimate of their effective EPSP rate, relative to the case of cortical pyramidal neurons, which have sparse firing and EPSP inputs (as well as greater diversity across neurons). We could add information about this in the main manuscript if the reviewer thinks it would be helpful.

2) Related to the first point, the choice of the 21 neuronal types considered (Suppl. Table 1 and Figure 8) and of their respective EPSP rates does not seem as straightforward as the authors may claim it. In my opinion, this is the most critical point of the current study, because the experimental validation of the theory totally relies on the EPSP rates affected to the different neuronal types where the biophysical properties of synaptic inhibition have been experimentally determined. Some values and choices are rather confusing. Based on a previous study (Fiorillo et al., 2014), the authors postulate that, in general (independent of neuronal types), the probability of firing of neurons in response to EPSPs can be assumed to be 50%. This is only partly true if we consider that i) much variability is present in each neuronal type, and ii) this probability is 100% in some cells and

only 25% in others. The experimental results arguing in favor of the 50% postulate have been mainly obtained from sensory nuclei (LGN and MNTB) (Fiorillo et al., 2014). Moreover, the Figure 7a shows that the predictive homeostasis theory gives a firing rate lower than 50% (between 20 and 40% for increasing rates of EPSGs) in most of the cases studied (except for learning rule 1 where IPSP decay rate is already set to its optimal value before learning). The authors should explain this discrepancy. The arguable choice of 50% for the relationship between spike rate and EPSG rate is critical, because for a number of the 21 neuronal types used in the current study, the authors extrapolate EPSG rates based only on the firing frequency of neurons (if a neuron fires at 10Hz, then we can assume that EPSG rate is the double, 20Hz). As said before, in a cell type where the probability of an EPSC to trigger a spike is 100% (or 25%), this extrapolation cannot be used. Moreover, if we take the specific example of monoaminergic and basal ganglia neurons (top of supplementary table 1), it is known that most of these neurons are spontaneously active in the absence of excitatory inputs (in slices for instance, case of SNc DA neurons, SNr GABAergic neurons, DR 5HT neurons, striatum ACh neurons). A large part of the firing activity of these neurons is due to their intrinsic properties and not to incoming glutamatergic inputs. To my knowledge, the 50% relationship between EPSG rate and firing frequency in these neurons is purely extrapolated from neuronal types that are mainly activated by synaptic inputs (such as thalamic sensory neurons or auditory nucleus neurons). The monoaminergic and basal ganglia neurons are particularly critical for the authors' conclusions, as they display some of the slowest EPSG rate (10Hz) of the cohort of neuronal types used for Figure 8. This class of neurons is just one example, but in general, the calculation/definition (and often extrapolation from sparse experimental data) of EPSG rates in the different neuronal types selected for the current study is rather vague and debatable (3 pages of the supplementary materials are necessary to justify these values). This point constitutes my main criticism of the article, as Figure 8 represents the strongest argument in favor of the theory proposed by the authors. If most of the values of EPSG rates are questionable, then the whole theory collapses or is at least strongly undermined.

There are three issues raised that I will address here, but the last sentence regarding estimation of EPSG rates is the most critical.

1. We devoted a lot of text (as well as statistics) to this key issue in the original manuscript, but we have expanded it in the revised manuscript. We argue that even an error of a few fold would be small relative to the large range of EPSG rates (100 fold) known to exist across our 21 neurons. However, our simple and most compelling argument is that our theory accurately predicted IPSPG decay times even if we completely disregard estimates of natural EPSG rates. Therefore our most basic conclusion does not depend on estimates of natural EPSG rates across neurons (to be precise, it still depends on an estimate of a typical or average EPSG rate, like 10 or 100 Hz, but the accuracy of an estimate of the mean or median rate across diverse types of neurons is not in serious doubt). Below is the relevant text from revised Results and Discussion.

Results

Estimation of natural EPSG rates is difficult for a variety of reasons. Although we provide detailed justification for our estimate in each of the 21 types of neuron (**Supplementary Methods**), uncertainty in these estimates limits confidence in the conclusions drawn from figure 10a,b. However, we would expect that the error in our estimates is approximately a factor of 2 (for example, if we estimate 20 Hz, the true rate is probably between 10 and 40 Hz), much less than the factor of 100 that distinguishes neurons with the fastest and slowest EPSG and spike rates (average EPSG rates were 3.7 Hz in locus coeruleus⁴², and >500 Hz in medial superior olive, even without acoustic stimulation⁴³). Therefore uncertainty in our estimates of EPSG rates does not substantially detract from our conclusion that theory can explain variability in IPSPG decay times across neurons.

However, we can entirely avoid the difficulty of estimating natural EPSG rates if we instead focus on the important but lesser challenge of explaining only the typical or average IPSPG decay time. In other words, why are IPSPG decay times on the order of 10 ms across many types of neurons? If we assume that 50 Hz is the median EPSG rate across real neurons, theory predicts optimal to have a median value of 5.1 ms (rule 2 in **Fig. 8a**), whereas the median across the 21 neurons was 7.0 ms. With this prediction of 5.1 ms, the error (RMSE) is a factor of 2.8, approximately twice that observed using EPSG rates as an explanatory factor across neurons (1.9; see above). Therefore the probability of observing this match between theory and experiment doubles, but is only 10^{-9} . Theory provides a remarkably accurate prediction of typical IPSPG decay times, even if we disregard estimates of EPSG rates.

Discussion:

•••• Our prediction was within a factor of 2 despite an infinite range of possible decay times. Although one could question our estimates of natural EPSG rates in the 21 neurons, the error in prediction increased only to a factor of 2.8 when we did not utilize these estimates as a predictive factor, which is still slightly more accurate than the factor of 3.2 for a prediction based on the mean decay time across the 21 neurons. Thus, if we disregard our estimates of natural EPSG rates (Fig. 10a,b), then we cannot claim to have explained variability in IPSP decay across neurons, but we can still claim to have explained the typical or average IPSP decay time.

We tried to emphasize that accurate prediction of the typical IPSP decay time is not trivial. It may seem trivial, because physiologists have measured it many times over many years, and we learn as students that it is near 10 ms. But the goal of theory is to predict it without using knowledge of data, and the range of possible decay times is infinite (a very slow decay time is indistinguishable from a leak conductance). Nonetheless, we predicted decay times across the 21 types of neuron within a factor of 2.8. We predicted a typical decay time of 5.1 ms (assuming a typical EPSG rate of 50 Hz), and the median across the 21 neurons was 7.0 ms.

2. Yes, many of the slow firing neuronal types do fire spontaneously in brain slices, even with neurotransmission blocked. These neurons are of particular importance here, since they are at one extreme of the range of EPSG rates. I am most familiar with the case of midbrain dopamine neurons, but other monoaminergic neurons have similar electrical properties. EPSCs occurred with an average rate of about 3 Hz in locus coeruleus of anesthetized animals (Sugiyama et al., 2012; which is now cited in the main manuscript). For other cell types we do not have direct evidence of EPSG rates. It is often assumed that because they fire spontaneously in brain slices, they are likely to do so in vivo. However, we have argued that almost every spike in dopamine neurons is caused by synaptic excitation in vivo (Fiorillo et al., 2014), as is known to occur in LGN. This is now stated explicitly in Supplementary Methods. I don't know of any direct evidence to prove or disprove this hypothesis, but it is known that rates of synaptic excitation and inhibition are much higher in vivo, and it is easy to imagine that synaptic conductance dominates neuronal

conductance, unlike the case of brain slices. It is well established that this is the case during “phasic” responses to reward stimuli.

3. In Supplementary Methods, we explained that to estimate EPSG rates in some neurons (including dopamine neurons), we assumed that EPSG rate was twice the average spike rate. We have a theoretical justification for this, as well as published data in some sensory neurons (Fiorillo et al., 2014). However, as we argue in Results, if we are off by a factor of 2 or even 3 in this estimate, it would not make a very large difference given the known range of 100 fold in EPSG rates (locus coeruleus at 3 Hz, medial superior olive over 500 Hz). More fundamentally, as we explain above, even if we disregard all estimates of EPSG rates, it would not change our central conclusion (in support of the theory).

A more tangential issue concerns the relation of long-term average spike probability in the present study (Fig. 9a) to theory (Fiorillo et al., 2014). We have added a paragraph on this issue to Supplementary Results (the second paragraph copied below).

Supplementary Results:

These differences in spike probability can be explained by considering the distributions of conductance, voltage, and residuals. At low EPSG frequencies, these distributions are highly skewed, with a mode at amplitudes below spike threshold and a ‘tail’ of higher amplitude events associated with spikes (**Supplementary Fig. 3**). As frequency increased, the distributions became more symmetric and their centers shifted closer to spike threshold. Qualitatively similar results were found for our log-normal distribution of unitary EPSG amplitudes (not shown). Spike-based rules (especially rule 1) tend to select IPSP parameters so that the median EPSP peak is near spike threshold, with the result that spike probability is close to 1/2. In contrast, minimizing MSR (distance from spike threshold) causes the mean EPSP peak to be near threshold. Because the distribution is skewed at low frequencies, this results in spike probability further below 1/2.

The central proposal of the theory tested here is that perfect homeostasis corresponds to the peak of every EPSP being exactly at spike threshold (Fiorillo et al., 2014). Although such perfection is not achievable, we previously reasoned that the long-term average spike probability should be near one half in neurons that are effective in maintaining homeostasis (Fiorillo et al., 2014). The present results demonstrate that the long-term spike probability predicted by theory to be present in neurons is not one half in general, but instead depends on the non-linear membrane properties and skewed distributions discussed above, which we did not consider previously. Since optimization of

homeostatic conductances causes the mean or median excitatory event to be near spike threshold, a consequence of such skewed distributions (Supplementary Fig. 3) is a long-term spike probability below one half (Fig. 9a).

Reviewer #3 (Remarks to the Author):

This manuscript explores the potential role of synaptic inhibition in counterbalancing synaptic excitation in what the authors describe as a "homeostatic balance" between excitation and inhibition. The computational studies appear well motivated, executed, and described. That said, I do not fully agree with the interpretation (and for that matter, the title of the manuscript) as it refers to homeostasis. However, I am not insisting on major terminology changes, but I just want to reiterate that "homeostasis", in my mind, is not what this paper is about. For other (mostly minor) comments, please refer to the attached Word document with tracked comments and suggested edits.

The most major comment concerns our use of the word "homeostasis." We have used this word in two previous titles (Fiorillo et al., 2014; Hong et al., 2014), and we believe that we are using it in accord with its standard usage. In our proposed model, neurons perform work (consume energy) to drive membrane excitability towards an optimal set point, or target, that we call "perfect homeostasis" (achieved when the peak of an EPSP is exactly at spike threshold). This is fully analogous to use of a balance scale for measuring weight, and it uses negative feedback. It unfolds through two types of mechanisms with different timescales. On the fast timescale, inhibition counteracts excitation. On a slower timescale, anti-Hebbian learning adjusts ion channel densities (GABA-A receptors) to keep EPSP peaks close to spike threshold. I

I do not agree with the reviewer that homeostasis necessarily refers to a slow process. However, anti-Hebbian plasticity is slow relative to the millisecond timescale of synaptic activity and spikes. "Distance from homeostasis" is defined in our model in real time, but to measure it we must consider the response to an EPSP. This is analogous to placing a "new weight" on a balance scale to assess how well it was predicted by the "reference weight."

The reviewer asked (in comments inserted into a copy of the manuscript) how inhibition can be said to "predict" or "anticipate" excitation (line 85). We have argued that there are two senses in which this word is appropriate

(Fiorillo et al., 2014). First, on a fast timescale, inhibition must “predict” or “estimate” excitation in order to counterbalance it. This is not entirely standard usage of “predict,” which I use as a synonym of “estimate” without reference to past or future. However, on a slower timescale, anti-Hebbian learning adjusts synaptic weights in order to achieve balance. Our use of “predict” (or anticipate) entirely consistent with standard usage, since the strength of synaptic inhibition (number of GABA-A receptors) must necessarily be established before synaptic excitation. The circuitry and biophysics of the brain must be set up in advance of the sensory events for which they are designed. Thus we can say that “brain structure predicts the structure of the sensory world.” I recognize that some people prefer to restrict the term “predict” to higher cognitive functions. But I find a more general definition to be more useful, and my use of the term is consistent with standard use with respect to “predictive coding” and “prediction error.”

Comment I am quite skeptical of this statement – I think that it is much more likely that different systems employ different operating principles, than that there is one mechanism that applies to all.

In my view, “prediction” is a general principle that applies to every cell. Certainly there are others who view ‘prediction’ as the general function of the brain, such as Karl Friston. The present manuscript provides evidence that one simple model can explain IPSP decay times, including variability across diverse neurons.

Although this is an important issue, the reviewer brought it up with respect to line 423:

Our intention was to mimic a typical neuron, and in principle such a generic “average model” may be the best single model for explaining variance in IPSP across diverse neurons.

This sentence was in the context of biophysical models of neurons, not “model” as in “theory.” The point is that a simple generic model with average properties, like our standard model, may be appropriate if one has to choose one biophysical model to explain variability across all neurons. Obviously more variability can be explained by making distinct biophysical models for each specific neuronal type (a multi-compartment model, for example). This is a minor issue that is distinct from the major issue of the generality of the theory to all types of neuron.

Our reference to “real EPSG” in the legend of figure 2 caused confusion. We have added text to the legend to clarify this (new text in blue).

Figure 2 Optimal inhibition in relation to EPSG interval. **(a)** Method of measuring ‘residuals.’ Top, two EPSG and the resulting EPSP. Dashed line indicates -50 mV, the “spike threshold” in these simulations without AP. Middle, after finding the voltage response, “test EPSG” (blue) were injected at the time of “real EPSG” (black) in order to find the “threshold EPSG” (red) that caused an EPSP with its peak to be nearest to threshold (in other words, EPSG amplitude was varied, but this “test variation” was only to find spike threshold, and had no influence on “real” voltage at the time of subsequent EPSG). Bottom, the residual was the difference in peak amplitudes between threshold and real EPSG. The residual depended on current and past but not future synaptic events; thus residual 1 in this example (left) was found in simulations in which EPSG 2 did not occur. **(b)** MSR as a function of the ratio of G_L to E (30 nS) for two EPSG separated by 5 (black), 10 (red), and 100 ms (blue) in the absence of synaptic inhibition ($I/E = 0$). **(c)** As in ‘b’ but showing MSR as a function of I/E ($=10$ ms, $G_L=0.1$ nS). **(d)** Heat plots of MSR as a function of both G_L/E and I/E for inter-EPSG intervals of 5 (left) and 10 ms (right). ‘X’ indicates minimum MSR.

Text in figure 3 and others was enlarged, and a couple of errors in identification of colors in the legends was correct, as suggested by the reviewer.

Reviewers' Comments:

Reviewer #2 (Remarks to the Author):

The authors made a real effort in answering my previous concerns, and I have no more major concern about the manuscript. However, I agree with reviewer 3 that the use of "homeostatic" in the current frame does not fit the usual definition of homeostatic plasticity, as described in the work of Gina Turrigiano, Graeme Davis, and others. Although homeostasis is a general term and an old concept, here it seems that the study is describing perfect excitation/inhibition balance rather than homeostatic plasticity. The central results of the paper, which does not incorporate homeostatic rules (changes in excitability or synaptic properties occurring over several hours), describe the possibility that, through the course of evolution, specific components of excitatory and inhibitory synapses may have been selected based on their dynamical complementarity, such that inhibition balances excitation, independent of the rate of synaptic events. Therefore, changing the title to "theory of optimal excitation/inhibition balance predicts..." would seem more appropriate, considering the usual definition of homeostatic plasticity used nowadays in neuroscience.

Reviewer #2 (Remarks to the Author):

The authors made a real effort in answering my previous concerns, and I have no more major concern about the manuscript. However, I agree with reviewer 3 that the use of "homeostatic" in the current frame does not fit the usual definition of homeostatic plasticity, as described in the work of Gina Turrigiano, Graeme Davis, and others. Although homeostasis is a general term and an old concept, here it seems that the study is describing perfect excitation/inhibition balance rather than homeostatic plasticity. The central results of the paper, which does not incorporate homeostatic rules (changes in excitability or synaptic properties occurring over several hours), describe the possibility that, through the course of evolution, specific components of excitatory and inhibitory synapses may have been selected based on their dynamical complementarity, such that inhibition balances excitation, independent of the rate of synaptic events. Therefore, changing the title to "theory of optimal excitation/inhibition balance predicts..." would seem more appropriate, considering the usual definition of homeostatic plasticity used nowadays in neuroscience.

Response to reviewer 2

We have removed the term "homeostatic" from the title and abstract. We introduce the term "predictive homeostasis" in Introduction, and we have added a sentence there to explain why we think this term is appropriate and aids understanding (see below). I gave our justification for 'homeostasis' in my previous response to reviewer 3, and I am not certain whether or not reviewer 2 read it. I am still not sure how the reviewers define 'homeostasis,' but I think our use is entirely consistent with the standard definition. I gave our definition in our previous response to reviewer 3.

Revised text in blue

Balance is difficult to achieve because EPSPs vary dynamically across a large range of amplitudes and on a timescale of a millisecond or less. Balance requires that inhibition is dynamically adjusted to predict EPSP amplitude, analogous to continually adjusting the reference weight in anticipation of a new weight being placed on the balance scale. We have therefore referred to this process as "predictive homeostasis"²¹, and it is related to principles of "predictive coding"²³⁻²⁵. It is "homeostatic" insofar as the parameters of inhibition (synaptic strength, decay time, etc.) are adjusted through negative feedback (e.g., anti-Hebbian plasticity) to drive membrane excitability towards an intermediate target (optimal balance), and it is "predictive" insofar as these parameters are determined in advance to counterbalance the expected future amplitude of excitation.